# Optimal kernel regression bounds
# under energy-bounded noise

**Amon Lahr**
ETH Zurich
amlahr@ethz.ch

**Johannes Köhler**[*]
ETH Zurich
jkoehle@ethz.ch

**Anna Scampicchio**[*]
ETH Zurich
ascampicc@ethz.ch

**Melanie N. Zeilinger**
ETH Zurich
mzeilinger@ethz.ch

## Abstract

Non-conservative uncertainty bounds are key for both assessing an estimation algorithm's accuracy and in view of downstream tasks, such as its deployment in safety-critical contexts. In this paper, we derive a tight, non-asymptotic uncertainty bound for kernel-based estimation, which can also handle correlated noise sequences. Its computation relies on a mild norm-boundedness assumption on the unknown function and the noise, returning the worst-case function realization within the hypothesis class at an arbitrary query input location. The value of this function is shown to be given in terms of the posterior mean and covariance of a Gaussian process for an optimal choice of the measurement noise covariance. By rigorously analyzing the proposed approach and comparing it with other results in the literature, we show its effectiveness in returning tight and easy-to-compute bounds for kernel-based estimates.

## 1 Introduction

Many problems in machine learning can be phrased in terms of estimating an unknown (continuous) function from a finite set of noisy data. A popular, non-parametric technique to perform such a task and return *point-wise* estimates is given by the class of kernel-based methods [Wahba, 1990; Schölkopf and Smola, 2001; Suykens et al., 2002; Shawe-Taylor and Cristianini, 2004; Steinwart and Christmann, 2008]. Complementing such estimates with non-conservative and non-asymptotic uncertainty bounds enables evaluating their reliability, for example, in view of deploying Bayesian optimization [Berkenkamp et al., 2023; Sui et al., 2018] or model-based reinforcement learning [Kuss and Rasmussen, 2003; Chua et al., 2018] to safety-critical systems.

Classical uncertainty bounds for kernel-based methods have been developed in statistical learning theory [Cucker and Smale, 2002; Cucker and Zhou, 2007; Guo and Zhou, 2013; Lecué and Mendelson, 2017; Ziemann and Tu, 2024]. However, these results are mostly aimed at characterizing the learning rate of the kernel-based algorithm and they tend to be difficult to apply in practice, being overly conservative or even depending on the unknown function to be estimated. Another viewpoint is given by Gaussian process (GP) regression [Rasmussen and Williams, 2006], a kernel-based method that is naturally endowed with an uncertainty quantification mechanism. However, closed-form uncertainty bounds are only available when assuming independent and Gaussian-distributed variables [Lederer et al., 2019], and their computation in other cases is non-trivial [Gilks et al., 1995]. To address this issue, high-probability and non-asymptotic uncertainty bounds have been derived by Srinivas et al. [2012]; Abbasi-Yadkori [2013]; Burnaev and Vovk [2014]; Fiedler et al. [2021]; Baggio et al. [2022]; Molodchyk et al. [2025], phrasing the problem as estimation in Reproducing Kernel Hilbert Spaces (RKHSs) [Aronszajn, 1950; Berlinet and Thomas-Agnan, 2004]. Yet, these bounds still heavily rely on the (conditional) independence of the noise sequence, which can be hard to satisfy in practice. This difficulty can be circumvented by leveraging an assumed bound on the

---

[*]Both co-authors contributed equally; their ordering is alphabetical.

39th Conference on Neural Information Processing Systems (NeurIPS 2025).

noise [Maddalena et al., 2021; Reed et al., 2025; Scharnhorst et al., 2023] – however, these results tend to be conservative or rely on solving a computationally intensive, constrained optimization problem to evaluate the uncertainty bound.

**Contribution** In this paper, we propose a novel non-asymptotic uncertainty bound for kernel-based estimation assuming a general bound on the noise energy. In particular, for each query input location, the proposed bound *exactly* characterizes the worst-case latent function within the given hypothesis class. The obtained uncertainty bound has the same structure as the high-probability uncertainty bounds from GP regression [Srinivas et al., 2012; Abbasi-Yadkori, 2013; Fiedler et al., 2021; Molodchyk et al., 2025], but with a measurement noise covariance $\sigma^2$ that depends on the test input location. Furthermore, we show that the derived bound recovers results from kernel interpolation [Weinberger and Golomb, 1959; Wendland, 2004] and linear regression [Fogel, 1979] as special cases. Finally, we contrast the proposed robust treatment to existing bounds for GP regression.

**Notation** The matrix $I_n \in \mathbb{R}^{n \times n}$ denotes the identity matrix of dimension $n$. For a symmetric positive-semidefinite matrix $A \in \mathbb{R}^{n \times n}$, $A^{1/2}$ denotes the (positive-semidefinite) symmetric matrix square root, i.e., $A^{1/2}A^{1/2} = A$, and $\|x\|_A^2 \doteq x^\top A x$ denotes the weighted Euclidean norm of a vector $x \in \mathbb{R}^n$. The Dirac delta function is denoted by $\delta : \mathbb{R}^{n_x} \to \mathbb{R}$, with $\delta(x) = 1$ for $x = 0$ and $\delta(x) = 0$ otherwise. Superscripts $f$ and $w$ will refer to the latent function and the noise, respectively. Accordingly, the kernel function is denoted by $k^\square : \mathbb{R}^{n_x} \times \mathbb{R}^{n_x} \to \mathbb{R}$, with $\square \in \{f, w\}$, and the associated RKHS is denoted by $\mathcal{H}_{k^\square}$. For two arbitrary ordered sets of indices $\mathbb{I}, \mathbb{J} \subseteq \mathbb{N}$, the matrix $K_{\mathbb{I}, \mathbb{J}}^\square \doteq [k^\square(x_i, x_j)]_{i \in \mathbb{I}, j \in \mathbb{J}}$ is the Gram matrix collecting the evaluations of the kernel function $k^\square$ at pairs of input locations $x_i, x_j$, with $i \in \mathbb{I}$ and $j \in \mathbb{J}$. We denote by $1 : N \doteq \{1, \dots, N\} \subseteq \mathbb{N}$ the set of indices for the training data points, while we use $x_{N+1}$ to represent the arbitrary test input. For instance, $K_{N+1, 1:N}^f \in \mathbb{R}^{1 \times N}$ corresponds to $\left[k^f(x_{N+1}, x_1), \dots, k^f(x_{N+1}, x_N)\right]$ and can be interpreted as the covariance matrix between the test- and training-input locations.

## 2 Problem set-up

We consider the problem of estimating an unknown latent function $f^{\text{tr}} : \mathcal{X} \to \mathbb{R}$, with $\mathcal{X} \subseteq \mathbb{R}^{n_x}$ from noisy measurements

$$y_i = f^{\text{tr}}(x_i) + w^{\text{tr}}(x_i), \quad i = 1, \dots, N, \tag{1}$$

collected at known training input locations $x_i \in \mathcal{X}$. Our goal is to compute worst-case uncertainty bounds around the latent function $f^{\text{tr}}$ given the observed data set $\mathcal{D} \doteq \{(x_i, y_i)\}_{i=1}^N$, which is subject to the unknown noise $w^{\text{tr}} : \mathcal{X} \to \mathbb{R}$. We phrase the problem in the framework of estimation in RKHSs [Aronszajn, 1950; Berlinet and Thomas-Agnan, 2004], and we model both the latent function $f^{\text{tr}}$ and the noise $w^{\text{tr}}$ as elements of an RKHS with a known kernel and a bound on their RKHS norm.

**Assumption 1.** The unknown latent and noise functions are respective elements of the RKHSs corresponding to the positive-semidefinite kernel $k^f : \mathcal{X} \times \mathcal{X} \to \mathbb{R}_{\geq 0}$ and the positive-definite kernel $k^w : \mathcal{X} \times \mathcal{X} \to \mathbb{R}_{\geq 0}$, where both $k^f$ and $k^w$ are uniformly bounded. There exist known constants $\Gamma_f$, $\Gamma_w > 0$ strictly bounding their respective RKHS norms, i.e., $\|f^{\text{tr}}\|_{\mathcal{H}_{k^f}}^2 < \Gamma_f^2$ and $\|w^{\text{tr}}\|_{\mathcal{H}_{k^w}}^2 < \Gamma_w^2$.

Characterizing boundedness of the noise using a kernel $k^w$ and an RKHS-norm bound $\Gamma_w^2$ provides a very general description and can model various scenarios: for instance, it captures the setting in which the noise sequence has bounded energy, as we elucidate in Section 4.1. Additionally, modeling noise as a deterministic quantity allows us to by-pass additional assumptions on the distribution or independence of the noise, latent function or input locations.

Since we model both the latent function and the noise as deterministic objects, there cannot be multiple output measurements at the same input location. Hence, we consider distinct inputs.

**Assumption 2.** The training input locations in $\mathbb{X} \doteq \{x_1, \dots x_N\}$ are pairwise distinct, i.e., $x_i \neq x_j$ for all $i, j = 1, \dots, N$ and $i \neq j$.

# 3 Kernel regression bounds for energy-bounded noise

In the following, we present the main result of our paper, determining tight point-wise uncertainty bounds $\underline{f}(x_{N+1}) \leq f^{\mathrm{tr}}(x_{N+1}) \leq \overline{f}(x_{N+1})$ for the value of the latent function at an arbitrary test point $x_{N+1} \in \mathcal{X}$. This task can be formulated as an infinite-dimensional optimization problem, taking the bounded-RKHS-norm assumption into account. The optimal upper bound $\overline{f}(x_{N+1})$ is defined as

$$\overline{f}(x_{N+1}) = \sup_{\substack{f \in \mathcal{H}_{k^f}, \\ w \in \mathcal{H}_{k^w}}} \quad f(x_{N+1}) \tag{2a}$$

$$\text{s.t.} \quad f(x_i) + w(x_i) = y_i, \; i = 1, \ldots, N, \tag{2b}$$

$$\|f\|^2_{\mathcal{H}_{k^f}} \leq \Gamma^2_f, \tag{2c}$$

$$\|w\|^2_{\mathcal{H}_{k^w}} \leq \Gamma^2_w. \tag{2d}$$

Analogously, the optimal lower bound is given by:

$$\underline{f}(x_{N+1}) = \inf_{\substack{f \in \mathcal{H}_{k^f}, \\ w \in \mathcal{H}_{k^w}}} \quad f(x_{N+1}) \tag{3a}$$

$$\text{s.t.} \quad \text{(2b)-(2d)}. \tag{3b}$$

In the following, we focus on computing the upper bound $\overline{f}(x_{N+1})$; for the lower bound, the presented results in Sections 3.1 and 3.2 are analogously derived in Appendices B and C, respectively.

Stated as in (2), the optimization problem is infinite-dimensional and is not directly tractable. Our key result, presented in the remainder of the section, consists in finding an exact reformulation of the constrained, infinite-dimensional problem (2) as a scalar, unconstrained one. The solution of the latter at an arbitrary input location is expressed in terms of familiar quantities from Gaussian process regression, for an optimal choice of measurement noise covariance. We present our derivation by first studying a relaxed formulation of this optimization problem in Section 3.1. Then, in Section 3.2, we discuss how to recover the optimal solution from the relaxed problem.

## 3.1 Relaxed solution

The relaxed formulation of optimization problem (2) considers the sum of the RKHS-norm constraints (2c), (2d) instead of enforcing them individually:

$$\overline{f}^{\sigma}(x_{N+1}) = \sup_{\substack{f \in \mathcal{H}_{k^f}, \\ w \in \mathcal{H}_{k^\sigma}}} \quad f(x_{N+1}) \tag{4a}$$

$$\text{s.t.} \quad f(x_i) + w(x_i) = y_i, \; i = 1, \ldots, N, \tag{4b}$$

$$\|f\|^2_{\mathcal{H}_{k^f}} + \|w\|^2_{\mathcal{H}_{k^\sigma}} \leq \Gamma^2_f + \frac{\Gamma^2_w}{\sigma^2}. \tag{4c}$$

This problem uses a scaled noise kernel $k^{\sigma}(x, x') \doteq \sigma^2 k^w(x, x')$ with a constant output scale $\sigma > 0$, which is key in relating the solution of the relaxed problem (4) to the original formulation (2). Additionally, note that the scaling implies $\|w\|^2_{\mathcal{H}_{k^\sigma}} = \|w\|^2_{\mathcal{H}_{k^w}}/\sigma^2 \leq \Gamma^2_w/\sigma^2$, as displayed in the constraint (4c).

The bound $\overline{f}^{\sigma}(x_{N+1})$ obtained from the relaxed problem depends on the noise parameter $\sigma$, as the joint RKHS-norm constraint (4c) is given by a weighted sum of both original constraints (2c) and (2d). Any feasible solution of (2) – a tuple $(f, w)$ of functions satisfying the constraints (2b)-(2d) – is also a feasible solution for Problem (4) for any $\sigma \in (0, \infty)$. Thus, $f^{\mathrm{tr}}(x_{N+1}) \leq \overline{f}(x_{N+1}) \leq \overline{f}^{\sigma}(x_{N+1})$, i.e., the uncertainty envelope obtained by solving the relaxed problem (4) contains the one obtained by solving the original problem (2) for all test points $x_{N+1} \in \mathcal{X}$ and noise parameters $\sigma \in (0, \infty)$.

Before stating the first result of this paper, the following definitions in terms of known quantities from Gaussian process regression are required. First, we define

$$f^{\mu}_{\sigma}(x_{N+1}) \doteq K^f_{N+1,1:N} \left( K^f_{1:N,1:N} + \sigma^2 K^w_{1:N,1:N} \right)^{-1} y, \tag{5a}$$

$$\Sigma^f_{\sigma}(x_{N+1}) \doteq K^f_{N+1,N+1} - K^f_{N+1,1:N} \left( K^f_{1:N,1:N} + \sigma^2 K^w_{1:N,1:N} \right)^{-1} K^f_{1:N,N+1}, \tag{5b}$$

with measurements $y \doteq [y_1, \ldots, y_N]^\top \in \mathbb{R}^N$. For the particular choice of noise kernel as $k^\sigma(x, x') = \sigma^2 k^w(x, x') = \sigma^2 \delta(x - x')$, it holds that $K^w_{1:N,1:N} = I_N$ and the above quantities respectively correspond to the GP posterior mean and covariance for independently and identically distributed (i.i.d.) Gaussian measurement noise with covariance $\sigma^2$ [Rasmussen and Williams, 2006, Chapter 2]. Additionally, we denote by

$$
\begin{aligned}
\|g_\sigma^\mu\|^2_{\mathcal{H}_{k^f + k^\sigma}} \doteq \min_{g \in \mathcal{H}_{k^f + k^\sigma}} \quad & \|g\|^2_{\mathcal{H}_{k^f + k^\sigma}} \\
\text{s.t.} \quad & g(x_i) = y_i, \ i = 1, \ldots, N, \\
= y^\top & \left( K^f_{1:N,1:N} + \sigma^2 K^w_{1:N,1:N} \right)^{-1} y
\end{aligned}
\tag{6}
$$

the RKHS norm of the minimum-norm interpolant in the RKHS $\mathcal{H}_{k^f + k^\sigma}$ defined for the sum of kernels $k^f + k^\sigma$ [Berlinet and Thomas-Agnan, 2004, Theorem 58]. Lastly,

$$
\beta_\sigma^2 \doteq \Gamma_f^2 + \frac{\Gamma_w^2}{\sigma^2} - \|g_\sigma^\mu\|^2_{\mathcal{H}_{k^f + k^\sigma}}
\tag{7}
$$

defines the maximum norm (4c) in the RKHS $\mathcal{H}_{k^f + k^\sigma}$ based on Assumption 1, reduced by the minimum norm required to interpolate the data.

The relaxed problem (4) admits the following closed-form analytical solution.

**Lemma 1.** *Let Assumptions 1 and 2 hold. Then, the solution of Problem* (4) *is given by*

$$
\overline{f}^\sigma(x_{N+1}) = f_\sigma^\mu(x_{N+1}) + \beta_\sigma \sqrt{\Sigma_\sigma^f(x_{N+1})}.
\tag{8}
$$

*Sketch of proof:* First, following arguments from the Representer Theorem [Kimeldorf and Wahba, 1971; Schölkopf et al., 2001], we show that the solution of Problem (4) is finite-dimensional. Next, two coordinate transformations are employed to reduce the number of free variables resulting from the interpolation constraint (4b), and to address the possible rank-deficiency of the kernel matrix $K^f_{1:N+1,1:N+1}$ for the latent function at the test and training input locations. Finally, the problem is reduced to an equivalent linear program with a norm-ball constraint that can be analytically solved. Expressing the solution in terms of the original coordinates then leads to (8). The detailed proof can be found in Appendix B.

This leads to the relaxed bound $\underline{f}^\sigma(x_{N+1}) \leq f(x_{N+1}) \leq \overline{f}^\sigma(x_{N+1})$, valid for all $\sigma \in (0, \infty)$. Due to the relaxation, the obtained upper and lower bounds are conservative with respect to the original problems (2) and (3) – nevertheless, the optimal solutions of (2), (3) can be retrieved for a suitable choice of the noise parameter $\sigma$, as shown in the following subsection.

## 3.2 Optimal solution

Our main result is formulated in the following theorem.

**Theorem 1.** *Let Assumptions 1 and 2 hold. Then, the solution of Problem* (2) *is given by*

$$
\overline{f}(x_{N+1}) = \inf_{\sigma \in (0, \infty)} \overline{f}^\sigma(x_{N+1}).
\tag{9}
$$

*Sketch of proof:* Similarly to Lemma 1, we first show that Problem (2) admits a finite-dimensional representation. The latter is analyzed depending on which of the constraints (2c) and (2d) are active, i.e., influence the optimal solution and have a corresponding strictly positive optimal Lagrange multiplier $\lambda^{f,\star}, \lambda^{w,\star}$. This leads to three non-trivial scenarios: In Case 1, it holds that $\lambda^{f,\star} > 0$, $\lambda^{w,\star} = 0$ and the optimal solution of (2) can be recovered by the relaxed problem for $\sigma^\star \to \infty$, for which the combined RKHS-norm constraint (4c) reduces to (2c). In Case 2, $\lambda^{f,\star} = 0$, $\lambda^{w,\star} > 0$ and $\sigma^\star \to 0$ recovers the optimal solution, rendering the constraint (4c) equivalent to (2d). In Case 3, both constraints are active and the optimal noise parameter is determined by $\sigma^\star = \sqrt{\lambda^{f,\star}/\lambda^{w,\star}} \in (0, \infty)$, i.e., the *ratio of the optimal Lagrange multipliers*. The set of active constraints at the optimal solution can be determined by case distinction, based on the feasibility of the primal solutions under the respective active-constraint set. Finally, it is shown that the optimal noise parameter $\sigma^\star$ in all three cases minimizes (9). This is illustrated in Fig. 1, which depicts the optimal noise parameters $\sigma^\star_{\text{sup}}, \sigma^\star_{\text{inf}}$,

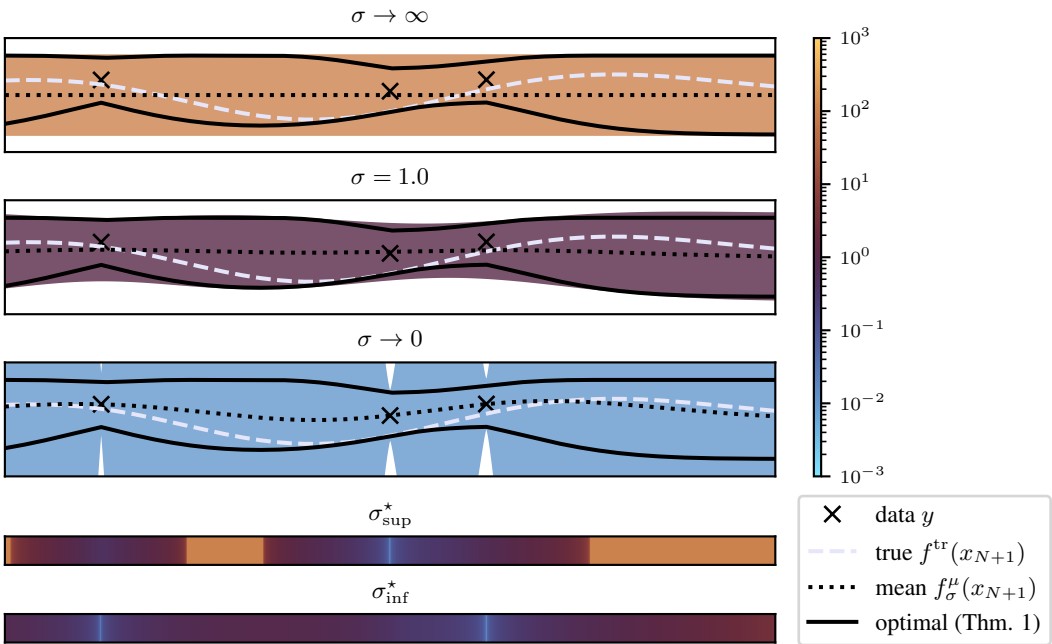

Figure 1: Illustrative example for Theorem 1. The optimal upper and lower bounds (solid black) for the (unknown) latent function $f^{\mathrm{tr}}$ (dashed white) are determined by the relaxed bounds (shaded) around the GP posterior mean (dotted black) for an optimal choice of noise parameter $\sigma^\star_{\mathrm{sup}}$ (upper bound) and $\sigma^\star_{\mathrm{inf}}$ (lower bound). The three upper plots show the relaxed upper and lower bounds, $\overline{f}^\sigma$ and $\underline{f}^\sigma$ for the values $\sigma = \{10^2, 10^0, 10^{-2}\}$, respectively. The two bottom colorbars indicate the respective optimal values $\sigma^\star_{\mathrm{sup}}$ and $\sigma^\star_{\mathrm{inf}}$ for the upper and lower bound. The plotted relaxed upper (lower) bounds equal the optimal upper (lower) bound for each test point where the color of the shaded area matches the color indicated in the colorbar for the optimal value $\sigma^\star_{\mathrm{sup}}$ ($\sigma^\star_{\mathrm{inf}}$).

for which the relaxed upper and lower bound, $\overline{f}^\sigma(x_{N+1})$ and $\underline{f}^\sigma(x_{N+1})$, correspond to the optimal bounds, $\overline{f}(x_{N+1})$ and $\underline{f}(x_{N+1})$, respectively. The detailed proof can be found in Appendix C.

Theorem 1 reduces the solution of the infinite-dimensional optimization problem (2) to a *scalar, unconstrained* optimization problem over the noise parameter $\sigma$. As such, it is amenable for efficient iterative optimization. Since running a fixed number of iterations of, e.g., gradient descent applied to Problem (9) returns a valid, improved upper bound $\overline{f}^\sigma(x_{N+1})$, this allows for iterative refinement of the uncertainty envelope. The solution thereby obtained can thus be easily integrated into existing pipelines for downstream tasks, such as uncertainty quantification in streaming-data settings or model-based reinforcement learning [Deisenroth and Rasmussen, 2011; Berkenkamp et al., 2017; Kamthe and Deisenroth, 2018].

### 3.3 Special cases

For both cases with only one active constraint, the optimal bound can be determined directly in closed form, without optimizing for the noise parameter $\sigma$. In the following, we provide the respective optimal solutions, as well as easy-to-evaluate expressions for determining the active constraint set. Noteworthy, the analytic solutions recover known bounds in specific regression settings, highlighting that the proposed bound is a generalization thereof; we detail these connections in Section 4.1.

**Case 1** ($\sigma \to \infty$). When the value $\Gamma^2_w$ is sufficiently permissive, constraint (2d) does not influence the optimal solution of (2). This leads to the optimal latent function $f^\star$ being chosen irrespective of the training data, while the optimal noise function $w^\star$ ensures consistency with the data (2b). The optimal bound $\overline{f}(x_{N+1})$ is then given by the prior GP covariance inflated by the full available RKHS norm $\Gamma_f$, recovering a classical kernel interpolation bound [Fasshauer and McCourt, 2015, Eq. (9.7)].

**Proposition 1.** *Let Assumptions 1 and 2 hold. If*

$$\left\| y - K_{1:N,N+1}^f \frac{\Gamma_f}{\sqrt{K_{N+1,N+1}^f}} \right\|_{(K_{1:N,1:N}^w)^{-1}}^2 \leq \Gamma_w^2, \tag{10}$$

*then the solution of* (2) *is given as*

$$\overline{f}(x_{N+1}) = \lim_{\sigma \to \infty} \overline{f}^\sigma(x_{N+1}) = \sqrt{K_{N+1,N+1}^f}\,\Gamma_f. \tag{11}$$

The feasibility condition (10) verifies if the bound (2d) on the noise function's RKHS norm allows for it to interpolate the points $w^\star(x_i) = y_i - f^\star(x_i)$, $i = 1, \ldots, N$, given the data-independent, worst-case latent function $f^\star(\cdot) = k^f(\cdot, x_{N+1}) \frac{\Gamma_f}{\sqrt{K_{N+1,N+1}^f}}$.

**Case 2** ($\sigma \to 0$). For infinite-dimensional hypothesis spaces, a regularity constraint on the latent function of the form (2c) is typically required to yield finite uncertainty bounds [Scharnhorst et al., 2023, Remark 1]. Therefore, it is possible that merely constraint (2d) is active only in degenerate cases – when the kernel matrix $K_{1:N+1,1:N+1}^f$ is singular, i.e., has rank $r \leq N$. The kernel matrix can then be expressed as $K_{1:N+1,1:N+1}^f = \Phi_{1:N+1}\Phi_{1:N+1}^\top$, where $\Phi_{1:N+1} \in \mathbb{R}^{(N+1)\times r}$ denotes the $r$-dimensional map of linearly independent features at the training and test input locations. This results in the following closed-form optimal solution of (2).

**Proposition 2.** *Let Assumptions 1 and 2 hold. Define $P \doteq (\Phi_{1:N}^\top (K_{1:N,1:N}^w)^{-1} \Phi_{1:N})^{-1}$ and $\theta^\mu \doteq P\Phi_{1:N}^\top (K_{1:N,1:N}^w)^{-1} y$. Then, if*

$$\|\theta^\star\|_2^2 \doteq \left\| \theta^\mu + \frac{P\Phi_{N+1}^\top}{\|\Phi_{N+1}^\top\|_P} \sqrt{\Gamma_w^2 - y^\top (K_{1:N,1:N}^w)^{-1} y + \|\theta^\mu\|_{P^{-1}}^2} \right\|_2^2 \leq \Gamma_f^2, \tag{12}$$

*the solution of* (2) *is given as*

$$\overline{f}(x_{N+1}) = \lim_{\sigma \to 0} \overline{f}^\sigma(x_{N+1}) = \Phi_{N+1}\theta^\mu + \|\Phi_{N+1}^\top\|_P \sqrt{\Gamma_w^2 - y^\top (K_{1:N,1:N}^w)^{-1} y + \|\theta^\mu\|_{P^{-1}}^2}. \tag{13}$$

Since the RKHS norm of the noise function is the limiting factor in this case, the optimal pair of functions $(f^\star, w^\star)$ generally shows the opposite behavior as in Case 1, utilizing the minimum RKHS norm of the noise $w^\star$ to interpolate the data in order to achieve a maximum value of the latent function $f^\star$ at the test point. The feasibility condition (12) verifies that the RKHS norm of the optimal latent function $f^\star$ satisfies the bound (2c).

Case 2 can happen in two scenarios: For *finite-dimensional* hypothesis spaces, i.e., $f^{\mathrm{tr}}(\cdot) = [\phi_1(\cdot) \ \cdots \ \phi_r(\cdot)] \theta^{\mathrm{tr}}$ for some features $\phi_i(\cdot)$, $i = 1, \ldots, r$, the latent function $f^\star(x_{N+1})$ generally does not have sufficient degrees of freedom to interpolate an arbitrary data set. As such, the optimal bound $\overline{f}(x_{N+1})$ in (13) consists of two components, the value of the *least-squares estimator* $f^\mu(x_{N+1}) \doteq \Phi_{N+1}\theta^\mu$, as well as a term proportional to the maximum RKHS norm of the noise $\Gamma_w^2$, subtracted by $y^\top (K_{1:N,1:N}^w)^{-1} y - \|\theta^\mu\|_{P^{-1}}$, the minimum RKHS norm required to eliminate the offset between the least-squares estimator and the data. For *infinite-dimensional* hypothesis spaces, the latent function $f^\star$ can generally interpolate the offset between the optimal noise function $w^\star$ and the training data; however, neglecting the RKHS-norm constraint (2c) on $f^\star$ only leads to sensible estimates when the test point coincides with a training input location. In this case, the feature vector $\Phi_{N+1} \in \mathbb{R}^{(N+1)\times r}$ has rank $N$, which simplifies the general result in Proposition 2.

**Corollary 1.** *Let Assumptions 1 and 2 hold. Suppose that $K_{1:N,1:N}^f$ is invertible and $x_{N+1} = x_k \in \mathbb{X}$. Then, if*

$$\|\theta^\star\|_2^2 \doteq \left\| y + K_{1:N,k}^w \frac{\Gamma_w}{\sqrt{K_{k,k}^w}} \right\|_{(K_{1:N,1:N}^f)^{-1}}^2 \leq \Gamma_f^2, \tag{14}$$

*the solution of* (2) *is given as*

$$\overline{f}(x_{N+1}) = \lim_{\sigma \to 0} \overline{f}^{\sigma}(x_{N+1}) = y_k + \sqrt{K_{k,k}^w} \Gamma_w. \tag{15}$$

In this case, the optimal solution at a training input location $x_{N+1} = x_k \in \mathbb{X}$ is given by the corresponding measurement $y_i$, inflated by the maximum RKHS norm $\Gamma_w$ of the noise function.

## 4 Related work and discussion

In this section, we discuss the obtained bounds also in view of known results from the literature. In particular, in Section 4.1 we detail known bounds that are recovered as special cases of Theorem 1. In Sections 4.2 and 4.3, we respectively compare Theorem 1 with *deterministic* bounds, obtained for bounded noise sequences, and *probabilistic* bounds, for noise sequences with an assumed probability distribution. Specifically, Section 4.3 encompasses a thorough numerical comparison of the bounds, including an application on safe control that shows the effectiveness of the proposed bounds on a downstream task.

### 4.1 Recovering existing bounds as particular cases

**Linear regression under energy-bounded noise**   We first elucidate the connection between Assumption 1 on the deterministic noise *function* and energy-boundedness of the noise *sequence*. As a straightforward consequence of the Representer Theorem [Wahba, 1990; Schölkopf et al., 2001], for the presented results in this paper, the values of the noise function outside the training input locations are irrelevant (see Appendix A): there exists a noise-generating function $w^{\mathrm{tr}}$ for the data set $\mathcal{D}$ satisfying Assumption 1 if and only if the minimum-norm interpolant $w^{\mu}$ of the (unknown) noise realizations $w_{\mathbb{X}} \doteq [w(x_1), \ldots, w(x_N)]^{\top}$ satisfies the RKHS-norm bound $\|w^{\mu}\|_{\mathcal{H}_{k^w}}^2 < \Gamma_w^2$, where

$$\begin{aligned}
\|w^{\mu}\|_{\mathcal{H}_{k^w}}^2 &\doteq \min_{w \in \mathcal{H}_{k^w}} \quad \|w\|_{\mathcal{H}_{k^w}}^2 \\
&\quad\quad \text{s.t.} \quad w(x_i) = w^{\mathrm{tr}}(x_i), \ i = 1, \ldots, N, \\
&= w_{\mathbb{X}}^{\top} \left( K_{1:N,1:N}^w \right)^{-1} w_{\mathbb{X}}.
\end{aligned}$$

Therefore, instead of imposing a maximum RKHS norm on $w^{\mathrm{tr}}$, one could equivalently assume a bounded RKHS norm for the minimum-norm interpolant generating the data set. For the Dirac noise kernel $k^w(x, x') = \delta(x - x')$, since $K_{1:N,1:N}^w = I_N$, the bounded-RKHS-norm assumption on the noise function implies bounded energy of the noise sequence, i.e.,

$$\|w^{\mu}\|_{\mathcal{H}_{k^w}}^2 = \sum_{i=1}^{N} w^{\mathrm{tr}}(x_i)^2 < \Gamma_w^2.$$

The assumption of bounded energy for the data set has been employed by Fogel [1979] to obtain bounds for the latent function in the setting of linear regression, i.e., finite-dimensional hypothesis spaces. Using the notation adopted in Section 3.3, the non-falsified parameter set is obtained as

$$\|\theta^{\mathrm{tr}} - \theta^{\mu}\|_{P^{-1}}^2 \le \Gamma_w^2 - y^{\top} y + \|\theta^{\mu}\|_{P^{-1}}^2, \tag{16}$$

see [Fogel, 1979, Eq. (3)]. Proposition 2 shows that the obtained bound recovers this known result from set-membership estimation for finite-dimensional hypothesis spaces. In fact, for the Dirac noise kernel, the worst-case realization of the unknown parameters is given by $\theta^{\star}$ in (12), for which (16) holds with equality. Note that the optimal bound in Theorem 1 does not only recover the bounds by Fogel [1979] for the linear-regression case, but, moreover, provides bounds under the additional complexity constraint $\|\theta^{\mathrm{tr}}\|_2^2 \le \Gamma_f^2$.

**Noise-free kernel interpolation**   Building upon the kernel interpolation bound by Weinberger and Golomb [1959] (see [Wendland, 2004, p. 192], [Fasshauer and McCourt, 2015, Section 9.3]), under Assumption 1 the following bound can be derived, cf. [Maddalena et al., 2021, Proposition 1]:

$$|f(x_{N+1}) - f_0^{\mu}(x_{N+1})| \le \sqrt{\Gamma_f^2 - \|f_0^{\mu}\|_{\mathcal{H}_{kf}}^2} \sqrt{\Sigma_0^f(x_{N+1})}, \tag{17}$$

where $f_0^\mu$ is the minimum-norm interpolant in the RKHS $\mathcal{H}_{k^f}$ (see (6)) and $\sqrt{\Sigma_0^f(x_{N+1})}$ is commonly referred to as the *power function*. The relaxed bound in Lemma 1 generalizes this result: for noise-free measurements, i.e., $\Gamma_w^2 \to 0$, and in the limit for the noise parameter $\sigma \to 0$, (17) is recovered exactly by noting that the bound in Lemma 1 is symmetric around the estimate.

## 4.2 Comparison with existing deterministic bounds

**Interpolation using sum-of-kernels**  For the Dirac noise kernel $k^w(x, x') = \delta(x - x')$, uncertainty bounds have also been obtained by [Kanagawa et al., 2025, Section 6.4] based on the minimum-norm interpolant $g_\sigma^\mu \in \mathcal{H}_{k^f + k^\sigma}$ using the sum of kernels $k^f + k^\sigma$. Utilizing the fact that, for all $x \notin \mathbb{X}$, the GP posterior mean (5a) and covariance (5b) are equal to the interpolant $g_\sigma^\mu$ and the corresponding power function, respectively, a bound on the true data-generating function $g \in \mathcal{H}_{k^f + k^\sigma}$ has been established by [Kanagawa et al., 2025, Corollary 6.8]. However, the bound does not take into account the actual value of the measurements $\{g(x_i) = y_i\}_{i=1}^N$, but rather the worst-case realization thereof, rendering it conservative. Additionally, the bound is only valid for the data-generating process $g$ and does not provide bounds for the latent function.

**Point-wise bounded noise**  As energy-boundedness is a weaker assumption than point-wise boundedness of the noise, Theorem 1 can also be applied in the setting of point-wise bounded noise, see Section 4.1. In this setting, [Maddalena et al., 2021; Reed et al., 2025] provide closed-form, yet conservative, bounds for the latent function under an RKHS-norm constraint on the latter. The bounds are improved upon by Scharnhorst et al. [2023], which provides optimal point-wise bounds for the latent function. As the bounded-energy and pointwise-boundedness assumptions are equivalent for $N = 1$ data points, so are the bounds by Scharnhorst et al. [2023] and Theorem 1 in this case. For larger data sets, under the point-wise-boundedness assumption, the optimal bounds by Scharnhorst et al. [2023] are tighter than the optimal bounds in Theorem 1 obtained under the weaker bounded-energy assumption. Still, their computation relies on solving a constrained convex program, cf. [Scharnhorst et al., 2023, Eq. (6)], whose number of optimization variables is proportional to the number of training data points $N$. Additionally. this optimization problem has to be solved to optimality in order to obtain valid bounds for the latent function, while optimization over the noise parameter $\sigma$ in (9) returns a valid upper bound for all $\sigma \in (0, \infty)$.

## 4.3 Comparison with existing probabilistic bounds

High-probability bounds for Gaussian-process regression are derived in [Srinivas et al., 2012; Abbasi-Yadkori, 2013; Fiedler et al., 2021; Molodchyk et al., 2025], which are generally of the form

$$\Pr\left[|f(x) - f_\sigma^\mu(x)| \le \beta_\sigma \sqrt{\Sigma_\sigma^f(x)}\ \forall x \in \mathcal{X}\right] \ge p.$$

Compared to Theorem 1, these bounds hold with a user-chosen probability $p \in (0, 1)$, use a fixed constant $\sigma > 0$, but otherwise have the same structure and the same assumption on the latent function $f^{\mathrm{tr}}$ in terms of a known bound on its RKHS norm. However, while the proposed analysis considers energy-bounded noise, $\|w^{\mathrm{tr}}\|_{\mathcal{H}_{k^w}}^2 < \Gamma_w^2$ (Assumption 1), these results apply to (conditionally) independent sub-Gaussian noise [Srinivas et al., 2012; Abbasi-Yadkori, 2013; Fiedler et al., 2021]. This is a stronger[1] requirement as it does not allow for biased or correlated noise, which can be difficult to ensure in real-world experiments. Nonetheless, both the proposed bound and existing high-probability bounds can be applied in case of independent, zero-mean and bounded noise; in the following, we numerically investigate the conservativeness of the bounds in this setting[2].

**Numerical comparison**  In this experiment, the size of the uncertainty regions is compared. Using a squared-exponential kernel $k^f(x, x') = \exp(-\|x - x'\|^2/\ell^2)$, $\ell = 1$, for the latent function, as well as a Dirac noise kernel $k^w(x, x') = \delta(x - x')$ on the domain $\mathcal{X} = [0, 4]$, random latent functions

---

[1]To be precise, if the noise is sub-Gaussian, we can derive an energy bound $\Gamma_w^2$, such that $w^{\mathrm{tr}}$ satisfies Assumption 1 with the kernel $k^w(x, x') = \delta(x - x')$ and a desired probability $p \in (0, 1)$. However, the converse is not true as energy-bounded noise may be correlated and biased.

[2]The code to reproduce the experiments is publicly available at `https://gitlab.ethz.ch/ics/bounded-energy-rkhs-bounds` and at `https://doi.org/10.3929/ethz-c-000785083`.

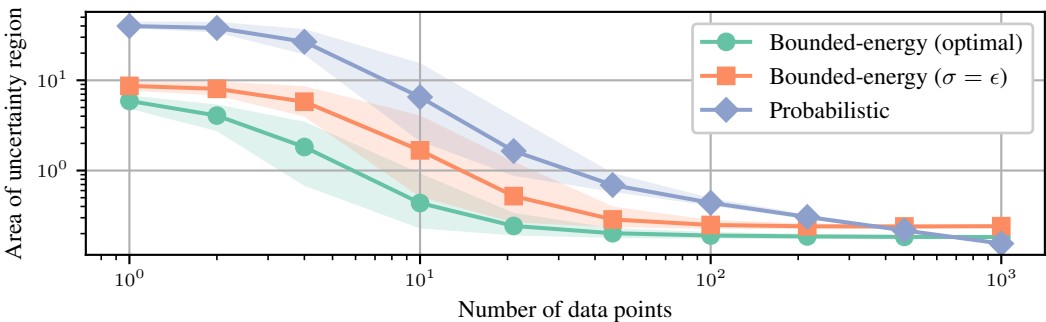

Figure 2: Numerical comparison of area of uncertainty region for increasing number of data points $N$ with $\{5\%, 95\%\}$-percentiles shown in shade.

are generated with $\|f^{\text{tr}}\|^2_{\mathcal{H}_{k_f}} = \Gamma^2_f = 1$. Training data are sampled based on measurement noise following a zero-mean truncated Gaussian distribution with standard deviation and bounded absolute value equal to $\epsilon = 0.01$, which is $R$-sub-Gaussian for $R = \epsilon$. The corresponding noise-energy bound is derived as $\Gamma^2_w = N\epsilon^2$. We compare the proposed bound (Theorem 1), which is optimal given only the information $\|w^{\text{tr}}\|^2_{\mathcal{H}_{k_w}} \leq \Gamma^2_w$, the relaxed bound (Lemma 1) with $\sigma = \epsilon$, and a standard high-probability error bound [Abbasi-Yadkori, 2013], cf. [Fiedler et al., 2024, Eq. (7)], which uses only sub-Gaussianity of the noise and provides a valid bound with probability $p = 0.99$, similar to [Srinivas et al., 2012; Fiedler et al., 2021]. Optimality of the numerical solution to (9) is guaranteed by solving a convex reformulation of (2) (see Appendix A) using CVXPY [Diamond and Boyd, 2016; Agrawal et al., 2018]. Figure 2 compares the area of the uncertainty region for $N = 1, \ldots, 10^3$ randomly sampled training points, averaged over $10^3$ runs for randomly sampled latent functions $f^{\text{tr}}$. In the low-data regime, the proposed optimal and relaxed bounds, leveraging energy-boundedness of the noise, are significantly less conservative. However, as multiple similar data points may provide no additional information without probabilistic information, they do not significantly improve after a certain number of data points $N$. In contrast, the probabilistic bound leverages independence, asymptotically attaining smaller uncertainty bounds with increasing data. However, it should be noted that these probabilistic bounds are only valid if indeed the noise is (conditionally) independent and zero-mean; otherwise, these shrinking confidence intervals would be misleading.

**Safe control for uncertain nonlinear systems** Lastly, we demonstrate the application of the proposed bounds to the downstream task of safe control. Consider the uncertain (for simplicity scalar) nonlinear dynamical system

$$x(k + 1) = f^{\text{known}}(x(k), u(k)) + f^{\text{tr}}(x(k), u(k)), \qquad (18)$$

with known dynamics $f^{\text{known}}$ and unknown residual dynamics $f^{\text{tr}}$. Given the current state $x(k) \in \mathbb{R}$ of the system at time $k \in \mathbb{N}$, the goal is to find an optimal control input $u(k) \in \mathbb{R}$ that minimizes a user-defined cost function $c(x(k), u(k))$, subject to a safety-critical constraint, $f^{\text{known}}(x(k), u(k)) + f^{\text{tr}}(x(k), u(k)) \geq (1 - \gamma)x(k)$ – similar to a control barrier function [Agrawal and Sreenath, 2017; Ames et al., 2019; Jagtap et al., 2020]. The uncertainty in the system dynamics is handled by leveraging the proposed (the probabilistic) bound to enforce *robust* constraint satisfaction for all functions in the uncertainty set, containing the unknown function $f^{\text{tr}}$ (with probability $p$). Importantly, this makes tight uncertainty bounds desirable, since they generally lead to lower costs and a larger feasible region, where safety of the control input can be guaranteed.

We compare the bounds for the following example setup: $f^{\text{known}}(x, u) \doteq 0.5x + u - 1$, $f^{\text{tr}}(x, u) \doteq \exp(-x^2)\sin(10x)$, $c(x, u) \doteq (f^{\text{known}}(x, u) + f^\mu_\sigma(x, u))^2 + u^2$, $k^f$, $k^w$ as above with $\ell = \sqrt{2}/20$. For the proposed bound, the optimization problem is formulated using the relaxed bound in Eq. (8), optimizing $\sigma \in (0, \infty)$ and $u(k) \in [-2, 2]$ *simultaneously*; for the probabilistic bounds, $\sigma = \epsilon$ is fixed with the same noise assumptions as in Section 4.3; see Appendix D for implementation details. Additionally, we implement the proposed bound using only the nearest 10 training points to construct the uncertainty bounds. Fig. 3 shows the success rate in terms of the share of feasible problems for an increasing amount of training data on a grid of 500 test points in the domain $x(k) \in [-2, 2]$, repeated 20 times with random noise realizations. Due to the smaller uncertainty bounds, the proposed bound

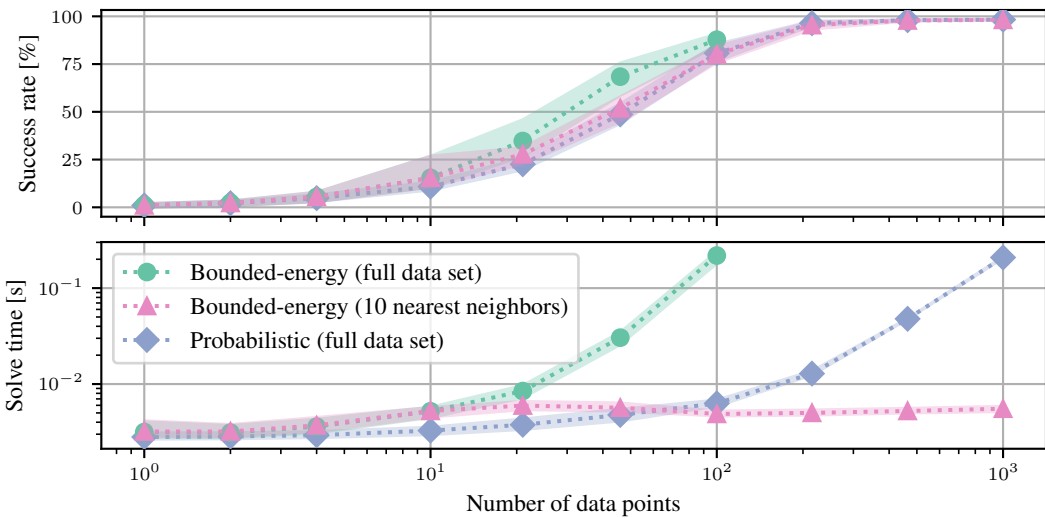

Figure 3: Application of uncertainty bounds for safe control. Success rate (upper plot) and solve time (lower plot) with $\{5\%, 95\%\}$-percentiles shown in shade.

using the full data set achieves the highest success rate, albeit at a high computational cost. In contrast, using the probabilistic bounds and the subset-of-data variant of the proposed bounds leads to similar success rates, with the latter exhibiting significantly lower computation times due to its independence of the number of training points. Note that utilizing the probabilistic bounds with test-point-dependent subsets of data would generally deteriorate the probability $p$ of their joint validity for all test points, compromising the controller's safety guarantees.

## 4.4 Limitations

The obtained bounds suffer from common criticalities and limitations of kernel-based learning, which are (a) dealing with kernel mis-specification and (b) knowing valid RKHS-norm bounds. Both issues are typically addressed empirically, (a) by hyper-parameter tuning via cross-validation or maximum-likelihood estimation ([Wahba, 1990], [Rasmussen and Williams, 2006, Section 5.4], [Karvonen et al., 2020]) and (b) by estimating the bound value from data [Csáji and Horváth, 2022; Tokmak et al., 2024]. While a rigorous investigation of (a) is beyond the scope of this paper, we note that mis-specification of $k^f$ may also be compensated by an inflated RKHS-norm bound $\Gamma_w$. Regarding (b), for the latent function, this is a common assumption in the literature on kernel-based uncertainty bounds; for the noise function, we discuss in Section 4.1 how it generalizes the common setting of energy-bounded noise. Under-estimation in the RKHS-norm bounds could be detected by checking feasibility of the optimization problem. Conversely, we point out that considering conservative values of $\Gamma_f$ and $\Gamma_w$ merely results in a sublinear inflation of the computed uncertainty envelope, cf. Eq. (7). Nevertheless, further research will be devoted to rigorously assessing the robustness of the obtained bounds with respect to possible mis-specifications in (a) and (b).

## 5 Conclusions

The main contribution of this paper is an optimization-based, *distribution-free* bound for kernel-based estimates that is tight, even in the non-asymptotic, low-data regime, and that can handle correlated and biased noise sequences. The proposed bound generalizes known results from kernel interpolation in the noise-free setting and from linear regression under energy-bounded noise. In the case of bounded sub-Gaussian noise, the numerical results highlight the competitiveness of the bound with existing probabilistic bounds in terms of its conservatism, and showcase its high potential for safe control as a downstream task. Moreover, the experiments highlight how the deterministic nature of the proposed bound enables the rigorous certification of subset-of-data selection or mixture-of-experts strategies to handle large data sets. Future work may investigate the effectiveness of the proposed bound for further downstream tasks, such as Bayesian optimization or model-based reinforcement learning.

## Acknowledgments and Disclosure of Funding

This work was supported by the European Union's Horizon 2020 research and innovation programme, Marie Skłodowska-Curie grant agreement No. 953348, ELO-X. The authors thank the anonymous reviewers for their constructive comments. AL thanks Philipp Hennig, Motonobu Kanagawa and Manish Prajapat for helpful discussions.

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

# Technical Appendix

The following sections contain the proofs of the mathematical claims made in the paper, as well as implementation details for the numerical examples. Specifically, Appendix A collects ancillary results, showing that the original infinite-dimensional problems yielding the upper- and lower bounds admit a finite-dimensional representation, which is the first step in computing their analytical solutions; additionally, it also presents two coordinate transformations that are useful for the following results. Appendix B provides the proof of Lemma 1. Appendix C contains the proof of Theorem 1, together with those for the special cases presented in Propositions 1 and 2 and Corollary 1. Finally, Appendix D provides further implementation details for the numerical example on "Safe control for uncertain nonlinear systems" in Section 4.3.

# A  Finite-dimensional representation of optimization problems

In this Section we first prove that optimization problems (2) and (4) admit a finite-dimensional representation (Lemma A.1 and Lemma A.2 in Appendix A.1). Next, in Appendix A.2 we present two coordinate transformations that will be deployed in the remaining sections.

## A.1  Representer Theorems

By using standard ideas from the representer theorem Kimeldorf and Wahba [1971]; Schölkopf et al. [2001] and [Scharnhorst et al., 2023, Appendix C.1], it can be established that the maximizer of (2) is finite-dimensional.

**Lemma A.1.** *A global maximizer of Problem* (2) *is given by*

$$f^\star(\cdot) = \sum_{i=1}^{N+1} k^f(\cdot, x_i) \alpha_i^f \in \mathcal{H}_{k^f}, \qquad w^\star(\cdot) = \sum_{i=1}^{N} k^w(\cdot, x_i) \alpha_i^w \in \mathcal{H}_{k^w}. \tag{A.1}$$

*Furthermore, Problem* (2) *is equivalent to the following finite-dimensional problem with $c^w \doteq K_{1:N,1:N}^w \alpha^w \in \mathbb{R}^N$:*

$$\overline{f}(x_{N+1}) = \sup_{\substack{\alpha^f \in \mathbb{R}^{N+1}, \\ c^w \in \mathbb{R}^N}} \quad K_{N+1,1:N+1}^f \alpha^f \tag{A.2a}$$

$$\text{s.t.} \quad K_{1:N,1:N+1}^f \alpha^f + c^w = y, \tag{A.2b}$$

$$(\alpha^f)^\top K_{1:N+1,1:N+1}^f \alpha^f - \Gamma_f^2 \le 0, \tag{A.2c}$$

$$\frac{1}{\sigma^2} \left( (c^w)^\top (K_{1:N,1:N}^w)^{-1} c^w - \Gamma_w^2 \right) \le 0. \tag{A.2d}$$

*Proof.* Let $\mathbb{X} = \{x_1, \dots, x_N\}$ be the set of training input locations and $\mathbb{X}_+ = \mathbb{X} \cup \{x_{N+1}\}$, the same set augmented with the test point. We denote by $\mathcal{H}_{k^f}^\| = \{f \in \mathcal{H}_{k^f} : f \in \mathrm{span}(k^f(\cdot, x_i), x_i \in \mathbb{X}_+)\}$ the span of kernel functions evaluated at the training and test input locations, as well as by $\mathcal{H}_{k^f}^\perp$ its orthogonal complement, i.e., $\mathcal{H}_{k^f}^\perp = \{f^\perp \in \mathcal{H}_{k^f} : \langle f^\perp, f^\| \rangle_{\mathcal{H}_{k^f}} = 0 \text{ for all } f^\| \in \mathcal{H}_{k^f}^\|\}$. Hence, any function $f \in \mathcal{H}_{k^f}$ can be written as $f = f^\| + f^\perp$, where $f^\| \in \mathcal{H}_{k^f}^\|$ and $f^\perp \in \mathcal{H}_{k^f}^\perp$. Note that the cost of the optimization problem is $f(x_{N+1}) = \langle f, k(x_{N+1}, \cdot) \rangle_{\mathcal{H}_{k^f}}$, which is insensitive to the orthogonal part $f^\perp$. Regarding the constraints, note that all functions $f^\perp \in \mathcal{H}_{k^f}^\perp$ do not affect the equality constraint (2b) while tightening the inequality constraints (2c); hence, it is optimal to set $f^\perp \equiv 0$. By the same arguments, it is optimal to set $w^\perp \equiv 0$. where the orthogonal complement is defined with respect to the finite-dimensional subspace $\mathcal{H}_{k^w}^\| = \{w \in \mathcal{H}_{k^w} : w \in \mathrm{span}(k^w(\cdot, x_i), x_i \in \mathbb{X})\}$, which excludes $k^w(\cdot, x_{N+1})$ as the cost is insensitive to $w(x_{N+1})$, the value of the noise function at the test point. Hence, it follows that for all functions $f \in \mathcal{H}_{k^f}$ and $w \in \mathcal{H}_{k^w}$, the respective orthogonal parts $f^\perp \in \mathcal{H}_{k^f}^\perp$ and $w^\perp \in \mathcal{H}_{k^w}^\perp$ can be set to zero without affecting feasibility or optimality of the candidate function.

Next, we show that the supremum is actually attained, i.e., that the optimizers $f^\star$ and $w^\star$ are elements of the respective finite-dimensional subspaces $\mathcal{H}_{k^f}^\|$ and $\mathcal{H}_{k^w}^\|$. First, we note that the norm constraints (2c) and (2d) define closed and bounded sets in the metric spaces $\mathcal{H}_{k^f}$ and $\mathcal{H}_{k^w}$, respectively. By the Cauchy-Schwartz inequality, the norm constraints (2c) and (2d) imply bounds on the pointwise evaluation of $f$ and $w$: $|f(x_i)| = \langle f, k^f(x_i, \cdot) \rangle_{\mathcal{H}_{k^f}} \le \|k^f(x_i, \cdot)\|_{\mathcal{H}_{k^f}} \|f\|_{\mathcal{H}_{k^f}} \le c_f \Gamma_f$, where $c_f \doteq \sup_{x \in \mathcal{X}} \sqrt{k^f(x, x)}$. Similarly, it holds that $|w(x_i)| = \langle w, k^w(x_i, \cdot) \rangle_{\mathcal{H}_{k^w}} \le \|k^w(x_i, \cdot)\|_{\mathcal{H}_{k^w}} \|w\|_{\mathcal{H}_{k^w}} \le c_w \Gamma_w$, where $c_w \doteq \sup_{x \in \mathcal{X}} \sqrt{k^w(x, x)}$. Note that $c_w, c_f < \infty$ holds by Assumption 1. Jointly with the data interpolation constraint (2b), this defines closed and bounded sets

$$D_i \doteq \left\{ (f(x_i), w(x_i)) \,\middle|\, f \in \mathcal{H}_{k^f}^\|, w \in \mathcal{H}_{k^w}^\|, \ f(x_i) + w(x_i) = y_i, |f(x_i)| \le c_f \Gamma_f, |w(x_i)| \le c_w \Gamma_w \right\}$$

in $\mathbb{R}^2$, for all $i = 1, \dots, N$. As the evaluation functionals $\mathbb{E}_{x_i}^f(f) = f(x_i) = \langle f, k^f(x_i, \cdot) \rangle_{\mathcal{H}_{k^f}}$ and $\mathbb{E}_{x_i}^w(w) = w(x_i) = \langle f, k^w(x_i, \cdot) \rangle_{\mathcal{H}_{k^w}}$ corresponding to the RKHSs $\mathcal{H}_{k^f}$ and $\mathcal{H}_{k^w}$, respectively, are

linear and continuous, the pre-image of $D_i$, $\mathrm{pre}(D_i) = \{(f, w) \mid (f(x_i), w(x_i)) \in D_i\}$, is closed in $\mathcal{H}_{k^f}^{\parallel} \times \mathcal{H}_{k^w}^{\parallel}$, for all $i = 1, \ldots, N$. Furthermore, the intersection of $\mathrm{pre}(D_i)$, $i = 1, \ldots, N$ and the bounded norm constraints (2c) and (2d) is closed and also bounded in $\mathcal{H}_{k^f}^{\parallel} \times \mathcal{H}_{k^w}^{\parallel}$ i.e., the feasible set of Problem (2) is closed and bounded. Since $\mathcal{H}_{k^f}^{\parallel}$ and $\mathcal{H}_{k^w}^{\parallel}$ are finite-dimensional, by the Heine-Borel theorem, the feasible set is compact; the value of the continuous objective $f(x_{N+1}) = \langle f, k^f(x_{N+1}, \cdot) \rangle_{\mathcal{H}_{k^f}}$ is thus attained by the Weierstrass extreme value theorem.

The finite-dimensional formulation (A.3) follows directly from inserting the finite-dimensional representations of $f^\star \in \mathcal{H}_{k^f}^{\parallel}$ and $w^\star \in \mathcal{H}_{k^w}^{\parallel}$ in (2) and defining $c^w = K_{1:N,1:N}^w \alpha^w$. $\qquad\square$

Similarly, we now prove that the relaxed infinite-dimensional problem (4) admits a finite-dimensional representation.

**Lemma A.2.** *A global maximizer of Problem* (4) *is given by* (A.1)*, and the resulting finite-dimensional problem can be written as*

$$\overline{f}^\sigma(x_{N+1}) = \sup_{\substack{\alpha^f \in \mathbb{R}^{N+1}, \\ c^w \in \mathbb{R}^N}} \quad K_{N+1,1:N+1}^f \alpha^f \tag{A.3a}$$

$$\text{s.t.} \quad K_{1:N,1:N+1}^f \alpha^f + c^w = y, \tag{A.3b}$$

$$\|\alpha^f\|_{K_{1:N+1,1:N+1}^f}^2 + \|c^w\|_{(\sigma^2 K_{1:N,1:N}^w)^{-1}}^2 \leq \Gamma_f^2 + \frac{\Gamma_w^2}{\sigma^2}. \tag{A.3c}$$

*Proof.* Analogous to Lemma A.1, it holds that setting $f^\perp \equiv 0$ retains optimality of any candidate function $f \in \mathcal{H}_k$, with $f = f^{\parallel} + f^\perp$. Similarly for the noise, it holds that $w^\perp \equiv 0$. Attainment of the supremum is also established along the lines of Lemma A.1, noting that the sum of norm-constraints (4c) defines a closed and bounded set in $\mathcal{H}_{k^f}^{\parallel} \times \mathcal{H}_{k^w}^{\parallel}$. Finally, the finite-dimensional optimization problem follows from replacing $f, w$ with their finite-dimensional expressions (A.1). $\quad\square$

### A.2 Coordinate transformations

We now present two transformations that will allow us to simplify the finite-dimensional representations (A.2) and (A.3). The first one will be used to deal with the possible rank-deficiency of the kernel matrix, the second one, to decompose the hypothesis space into orthogonal features. A subset of the corresponding weights will be fully determined by the training data, while the remaining ones will be adversarially chosen to obtain the worst-case value of the latent function at the test point. We point the interested reader to Müller and Schaback [2009]; Pazouki and Schaback [2011] for details on similar basis transformations for kernel spaces.

**Eliminating the null space of the kernel matrix**

For degenerate kernel functions, i.e., finite-dimensional hypothesis spaces, as well as in the case when the test point coincides with a training data point, the kernel matrix $K_{1:N+1,1:N+1}^f$ associated with the latent function $f$ can be singular. To handle the rank-deficiency, let us denote the rank of the matrix $K_{1:N+1,1:N+1}^f$ by $r$, which satisfies $r \leq N + 1$ by definition. To eliminate redundant variables, we employ a singular value decomposition (SVD) of $K_{1:N+1,1:N+1}^f$:

$$K_{1:N+1,1:N+1}^f = \begin{bmatrix} K_{1:N,1:N}^f & K_{1:N,N+1}^f \\ K_{N+1,1:N}^f & K_{N+1,N+1}^f \end{bmatrix} = VSV^\top \tag{A.4}$$

$$= \begin{bmatrix} V_{11} & V_{12} \\ V_{21} & V_{22} \end{bmatrix} \begin{bmatrix} S_r & 0 \\ 0 & 0 \end{bmatrix} \begin{bmatrix} V_{11}^\top & V_{21}^\top \\ V_{12}^\top & V_{22}^\top \end{bmatrix} \tag{A.5}$$

Thereby we have partitioned the rows of the orthonormal matrix $V$, with $VV^\top = I$, according to the separation of $K_{1:N+1,1:N+1}^f$ into evaluations at the training points and the test point. Note that if $K_{1:N+1,1:N+1}^f$ has full rank, i.e., $r = N + 1$, then the diagonal and positive-definite matrix $S_r$,

containing the non-zero singular values of $K^f_{1:N+1,1:N+1}$, is equal to the matrix $S$ containing all singular values, $S = S_r$; and the the matrices $V_{12}, V_{22}$ are void in this case. The first $r$ vectors in $V$ form a basis for the image of $K^f_{1:N+1,1:N+1}$: A coordinate transformation

$$\begin{bmatrix} v_1 \\ v_2 \end{bmatrix} = \begin{bmatrix} V_{11}^\top & V_{21}^\top \\ V_{12}^\top & V_{22}^\top \end{bmatrix} \alpha^f \qquad \Leftrightarrow \qquad \alpha^f = \begin{bmatrix} V_{11} & V_{12} \\ V_{21} & V_{22} \end{bmatrix} \begin{bmatrix} v_1 \\ v_2 \end{bmatrix} \tag{A.6}$$

reveals that $K^f_{1:N+1,1:N+1} \alpha^f = \begin{bmatrix} V_{11} \\ V_{21} \end{bmatrix} S_r v_1$. Hence, neither the optimal cost nor the constraints of (A.1) and (A.3) depend on $v_2$, implying that there exists an optimal solution which satisfies $v_2 = 0$. To simplify notation, we denote by

$$\Phi_{1:N+1} = \begin{bmatrix} \Phi_{1:N} \\ \Phi_{N+1} \end{bmatrix} = \begin{bmatrix} V_{11} \\ V_{21} \end{bmatrix} S_r^{1/2} \tag{A.7}$$

the *feature matrix* associated with the kernel matrix

$$K^f_{1:N+1,1:N+1} = \Phi_{1:N+1} \Phi_{1:N+1}^\top \tag{A.8}$$

Defining as $\theta \doteq S_r^{1/2} v_1 \in \mathbb{R}^r$ the corresponding weight vector, it holds that

$$K^f_{1:N+1,1:N+1} \alpha^f = \Phi_{1:N+1} \theta. \tag{A.9}$$

**Eliminating the subspace determined by training data**

Interpolation of the training data by the latent function and noise process uniquely determines the components of the optimal solution in an $N$-dimensional subspace, while the remaining orthogonal components are not affected by this constraint. We find this subspace by applying a QR decomposition

$$\begin{bmatrix} \Phi_{1:N}^\top \\ (\sigma R^w_{1:N,1:N})^\top \end{bmatrix} = \underbrace{\begin{bmatrix} Q_{11} & Q_{12} \\ Q_{21} & Q_{22} \end{bmatrix}}_{\doteq Q} \begin{bmatrix} R \\ 0 \end{bmatrix}, \tag{A.10}$$

where $Q \in \mathbb{R}^{(N+r) \times (N+r)}$ is an orthonormal matrix, $R \in \mathbb{R}^{N \times N}$ is upper-triangular, and

$$\sigma^2 K^w_{1:N,1:N} = \sigma R^w_{1:N,1:N} (\sigma R^w_{1:N,1:N})^\top \tag{A.11}$$

is the (upper-triangular) Cholesky decomposition of the noise covariance matrix or, equivalently,

$$(\sigma^2 K^w_{1:N})^{-1} = \left( \sigma R^w_{1:N,1:N} (\sigma R^w_{1:N,1:N})^\top \right)^{-1} = (\sigma R^w_{1:N,1:N})^{-\top} (\sigma R^w_{1:N,1:N})^{-1} \tag{A.12}$$

is the standard (lower-triangular) Cholesky decomposition of the inverse noise covariance matrix. We use the orthogonal matrix $Q$ from the QR decomposition to define a coordinate transformation

$$\begin{bmatrix} Q_{11} & Q_{12} \\ Q_{21} & Q_{22} \end{bmatrix} \begin{bmatrix} \delta_1 \\ \delta_2 \end{bmatrix} = \begin{bmatrix} I_r & 0 \\ 0 & (\sigma R^w_{1:N,1:N})^{-1} \end{bmatrix} \begin{bmatrix} \theta \\ c_w \end{bmatrix}, \tag{A.13}$$

with $\delta_1 \in \mathbb{R}^N$, $\delta_2 \in \mathbb{R}^r$, which allows compute the components of the solution determined by the training data. For clarity, we emphasize that the partitioning of the matrices in Eq. (A.13) on the left- and right-hand side is different: the first line on the left-hand side contains $N$ rows, the first line on the right-hand side, $r$ rows.

# B    Proof of Lemma 1

Starting from the finite-dimensional formulation of the relaxed problem (4) as given in (A.3), we first apply the two coordinate transformations presented in Appendix A.2 (Appendix B.1). This allows us to obtain a simplified problem formulation — a linear program with a single norm-ball constraint — that can be solved directly (Appendix B.2). Finally, we also present the result for the lower bound (Appendix B.3).

## B.1    Preliminary coordinate transformation

Using the SVD of the kernel matrix (A.5) as well as the coordinate transformation (A.13), the data equation (A.3b) reads

$$
\begin{aligned}
y &\overset{(A.9)}{=} \Phi_{1:N}\theta + c^w, \\
&= \begin{bmatrix} \Phi_{1:N} & \sigma R^w_{1:N,1:N} \end{bmatrix} \begin{bmatrix} I_N & 0 \\ 0 & (\sigma R^w_{1:N,1:N})^{-1} \end{bmatrix} \begin{bmatrix} \theta \\ c_w \end{bmatrix}, \\
&\overset{(A.10)}{=} \begin{bmatrix} R^\top & 0 \end{bmatrix} \begin{bmatrix} Q_{11}^\top & Q_{21}^\top \\ Q_{12}^\top & Q_{22}^\top \end{bmatrix} \begin{bmatrix} I_N & 0 \\ 0 & (\sigma R^w_{1:N,1:N})^{-1} \end{bmatrix} \begin{bmatrix} \theta \\ c_w \end{bmatrix} \\
&\overset{(A.13)}{=} R^\top \delta_1.
\end{aligned}
\tag{B.1}
$$

This leads to $\delta_1^{\star,\sigma} = R^{-\top}y$ being fully determined by the data, leaving only $\delta_2 \in \mathbb{R}^r$ to be optimized. The RKHS-norm constraint (A.3c) is reformulated as

$$
(\alpha^f)^\top K^f_{1:N+1,1:N+1}\alpha^f + (c^w)^\top(\sigma^2 K^w_{1:N,1:N})^{-1}c^w \overset{(A.8),(A.9)}{=} \left\| \begin{bmatrix} I_N & 0 \\ 0 & (\sigma R^w_{1:N,1:N})^{-1} \end{bmatrix} \begin{bmatrix} \theta \\ c_w \end{bmatrix} \right\|_2^2
$$

$$
\overset{(A.13)}{=} \|\delta_1\|_2^2 + \|\delta_2\|_2^2,
\tag{B.2}
$$

where we have used that $Q$ is orthogonal, i.e., $\|Qx\|_2^2 = \|x\|_2^2$ for all $x \in \mathbb{R}^{r+N}$. Finally, in the new coordinates, the cost is expressed as

$$
K^f_{N+1,1:N}\alpha^f \overset{(A.9)}{=} \Phi_{N+1}\theta \overset{(A.13)}{=} \Phi_{N+1}(Q_{11}\delta_1 + Q_{12}\delta_2).
\tag{B.3}
$$

Problem (A.3) is thus equivalently reformulated as follows:

$$
\overline{f}^\sigma(x_{N+1}) = \sup_{\delta_2 \in \mathbb{R}^r} \quad \Phi_{N+1}Q_{12}\delta_2 + \Phi_{N+1}Q_{11}\delta_1^{\star,\sigma}
\tag{B.4a}
$$

$$
\text{s.t.} \quad \|\delta_2\|_2^2 \leq \Gamma_f^2 + \frac{\Gamma_w^2}{\sigma^2} - \|\delta_1^{\star,\sigma}\|_2^2.
\tag{B.4b}
$$

## B.2    Analytical solution

With its linear cost and norm-ball constraint, problem (B.4) has the unique optimal solution

$$
\delta_2^{\star,\sigma} = \frac{Q_{12}^\top \Phi_{N+1}^\top}{\|Q_{12}^\top \Phi_{N+1}^\top\|_2} \sqrt{\Gamma_f^2 + \frac{\Gamma_w^2}{\sigma^2} - \|\delta_1^{\star,\sigma}\|_2^2}
\tag{B.5}
$$

and associated optimal cost

$$
\overline{f}^\sigma(x_{N+1}) = \Phi_{1:N}Q_{11}\delta_1^{\star,\sigma} + \|Q_{12}^\top \Phi_{N+1}^\top\|_2 \sqrt{\Gamma_f^2 + \frac{\Gamma_w^2}{\sigma^2} - \|\delta_1^{\star,\sigma}\|_2^2}
\tag{B.6}
$$

To obtain the formulation in in (8), we use the following relations inferred from the QR decomposition (A.10):

$$
Q_{12}Q_{12}^\top = I - Q_{11}Q_{11}^\top,
\tag{B.7a}
$$

$$
Q_{11} = \Phi_{1:N}^\top R^{-1},
\tag{B.7b}
$$

$$
R^\top R = R^\top Q^\top QR = \Phi_{1:N}\Phi_{1:N}^\top + (\sigma R^w_{1:N,1:N})^\top \sigma R^w_{1:N,1:N}
$$

$$= K^f_{1:N,1:N} + \sigma^2 K^w_{1:N,1:N}. \tag{B.7c}$$

We can now simplify the terms in the optimal cost (B.6). First, it holds that

$$\Phi_{N+1} Q_{11} \delta_1^{\star,\sigma} \overset{\text{(B.7b)}}{=} (\Phi_{N+1} \Phi_{1:N}^\top) R^{-1} R^{-\top} y,$$

$$\overset{\text{(B.7c)}}{=} K^f_{N+1,1:N} \left( K^f_{1:N,1:N} + \sigma^2 K^w_{1:N,1:N} \right)^{-1} y,$$

$$\overset{\text{(5a)}}{=} f^\mu_\sigma(x_{N+1}).$$

Then, we have

$$\|Q_{12}^\top \Phi_{N+1}^\top\|_2^2 = \Phi_{N+1} Q_{12} Q_{12}^\top \Phi_{N+1}^\top$$

$$\overset{\text{(B.7a)}}{=} \Phi_{N+1}(I - Q_{11} Q_{11}^\top) \Phi_{N+1}^\top$$

$$\overset{\text{(B.7b)}}{=} \Phi_{N+1} \Phi_{N+1}^\top - (\Phi_{N+1} \Phi_{1:N}^\top) R^{-1} R^{-\top} (\Phi_{1:N} \Phi_{N+1}^\top)$$

$$\overset{\text{(B.7c)}}{=} K^f_{N+1,N+1} - K^f_{N+1,1:N} \left( K^f_{1:N,1:N} + \sigma^2 K^w_{1:N,1:N} \right)^{-1} K^f_{1:N,N+1}$$

$$\overset{\text{(5b)}}{=} \Sigma^f_\sigma(x_{N+1}).$$

Lastly, we obtain

$$\|\delta_1^{\star,\sigma}\|_2^2 \overset{\text{(B.1)}}{=} y^\top R^{-1} R^{-\top} y$$

$$\overset{\text{(B.7c)}}{=} y^\top \left( K^f_{1:N,1:N} + \sigma^2 K^w_{1:N,1:N} \right)^{-1} y$$

$$\overset{\text{(6)}}{=} \|g^\mu_\sigma\|^2_{\mathcal{H}_{k^f+k^\sigma}}.$$

To summarize, this shows that the optimal cost of (4) is given by

$$\overline{f}^\sigma(x_{N+1}) = f^\mu_\sigma(x_{N+1}) + \sqrt{\Gamma_f^2 + \frac{\Gamma_w^2}{\sigma^2} - \|g^\mu_\sigma\|^2_{\mathcal{H}_{k^f+k^\sigma}}} \sqrt{\Sigma^f_\sigma(x_{N+1})}.$$

### B.3  Optimal relaxed solution for the lower bound

For the lower bound, the same derivations apply with a minor change. Flipping the sign in the cost leads leads to a flipped sign in the optimal solution for the free variables $\delta_2$, i.e., $\delta_2^{\star,\text{inf}} = -\delta_2^{\star,\sigma}$. This results in the optimal cost for the lower bound

$$\underline{f}^\sigma(x_{N+1}) = f^\mu_\sigma(x_{N+1}) - \sqrt{\Gamma_f^2 + \frac{\Gamma_w^2}{\sigma^2} - \|g^\mu_\sigma\|^2_{\mathcal{H}_{k^f+k^\sigma}}} \sqrt{\Sigma^f_\sigma(x_{N+1})}.$$

Due to the symmetry of the relaxed bounds around $f^\mu_\sigma(x_{N+1})$, the following corollary is immediate.

**Corollary 2.** *Let Assumptions 1 and 2 be satisfied. Then, for all $\sigma \in (0, \infty)$, it holds that*

$$|f^{\text{tr}}(x_{N+1}) - f^\mu_\sigma(x_{N+1})| \leq \sqrt{\Gamma_f^2 + \frac{\Gamma_w^2}{\sigma^2} - \|g^\mu_\sigma\|^2_{\mathcal{H}_{k^f+k^\sigma}}} \sqrt{\Sigma^f_\sigma(x_{N+1})}.$$

## C  Proof of Theorem 1

In the following, we derive an analytic solution to Problem (2). Taking its finite-dimensional formulation (A.2), we eliminate the noise coefficients as a function of the latent function coefficients, $c^w = y - \Phi_{1:N}\theta$, and deploy (A.7) to obtain the following reformulation:

$$\overline{f}(x_{N+1}) = \sup_{\theta \in \mathbb{R}^r} \quad \Phi_{N+1}\theta \tag{C.1a}$$

$$\text{s.t.} \quad \theta^\top \theta - \Gamma_f^2 \leq 0, \tag{C.1b}$$

$$(y - \Phi_{1:N}\theta)^\top (K^w_{1:N,1:N})^{-1}(y - \Phi_{1:N}\theta) - \Gamma_w^2 \leq 0. \tag{C.1c}$$

We will analyze the solution of Problem (C.1) for different active sets. Here, the term "active set" refers to a subset of the RKHS-norm constraints (C.1b) and (C.1c) that are strictly active, i.e., influence the optimal primal solution of the problem. For strictly active constraints (C.1b) and (C.1c) there exist respective Lagrangian multipliers, $\lambda^f$ and $\lambda^w$, that are strictly positive. We investigate the following combinations:

Case 1 : only (C.1b) is strictly active ($\lambda^f > 0$, $\lambda^w = 0$),

Case 2 : only (C.1c) is strictly active ($\lambda^f = 0$, $\lambda^w > 0$),

Case 3 : both (C.1b) and (C.1c) are strictly active ($\lambda^f > 0$, $\lambda^w > 0$).

Case 4 : both (C.1b) and (C.1c) are not strictly active ($\lambda^f = 0$, $\lambda^w = 0$),

Based on the solutions for fixed active sets[3], the optimal solution can then be found by case distinction. We discuss each case separately, obtaining the corresponding analytical solution, presenting the feasibility check and elucidating the connection with the solution of the relaxed problem in Appendices C.1 to C.4; note that Appendices C.2 and C.3 provide the proofs for Propositions 1 and 2 and Corollary 1. We then show how to practically check which set is active, and obtain the desired claim (9) in Appendix C.5. We conclude the section by presenting the result for the lower bound (Appendix C.6).

### C.1  Case 1: Noise constraint inactive

We now consider the case in which only (C.1b) is active, proving Proposition 1.

**Optimal solution**  Problem (C.1) with omitted constraint (C.1c) is given by

$$\overline{f}_1(x_{N+1}) = \sup_{\theta \in \mathbb{R}^r} \quad \Phi_{N+1}\theta \tag{C.2a}$$

$$\text{s.t.} \quad \theta^\top \theta - \Gamma_f^2 \leq 0. \tag{C.2b}$$

The solution of the above optimization problem is given as

$$\theta^{\star,1} = \frac{\Phi_{N+1}^\top}{\|\Phi_{N+1}\|_2}\Gamma_f, \tag{C.3}$$

which results in the optimal cost given in (11), namely

$$\overline{f}_1(x_{N+1}) = \frac{\Phi_{N+1}\Phi_{N+1}^\top}{\|\Phi_{N+1}\|_2}\Gamma_f \overset{(A.7)}{=} \sqrt{K^f_{N+1,N+1}}\Gamma_f.$$

**Feasibility check**  The optimizer $\theta^{\star,1}$ is a feasible solution of (C.1) if the corresponding optimal noise coefficients

$$c^{w,\star,1} \doteq y - \frac{\Phi_{1:N}\Phi_{N+1}^\top}{\sqrt{\Phi_{N+1}\Phi_{N+1}^\top}}\Gamma_f \overset{(A.7)}{=} y - K^f_{1:N,N+1}\frac{\Gamma_f}{\sqrt{K^f_{N+1,N+1}}},$$

---

[3]The presented analysis of Cases 1-3 considers a slightly more permissive setting, which allows some of the Lagrange multipliers to be zero, i.e., the corresponding constraint to be inactive or weakly active. Thus, in some scenarios multiple cases might be applicable (see Appendix C.5); yet, this does not affect our analysis as the cases cover all possible scenarios.

satisfy the neglected constraint (C.1c), i.e., if

$$\left\| y - K^f_{1:N,N+1} \frac{\Gamma_f}{\sqrt{K^f_{N+1,N+1}}} \right\|^2_{(K^w_{1:N,1:N})^{-1}} \le \Gamma^2_w,$$

as given in (15)).

**Connection to relaxed solution**  For $\sigma \to \infty$, for the relaxed solution $\overline{f}^\sigma(x_{N+1})$ in Appendix B, it holds that

$$\lim_{\sigma \to \infty} f^\mu_\sigma = \lim_{\sigma \to \infty} K^f_{N+1,1:N} \left( K^f_{1:N,1:N} + \sigma^2 K^w_{1:N,1:N} \right)^{-1} y$$

$$= 0,$$

$$\lim_{\sigma \to \infty} \beta_\sigma = \lim_{\sigma \to \infty} \sqrt{\Gamma^2_f + \frac{\Gamma^2_w}{\sigma^2} - \|g^\mu_\sigma\|^2_{\mathcal{H}_{kf+k\sigma}}}$$

$$= \Gamma_f,$$

$$\lim_{\sigma \to \infty} \Sigma^f_\sigma(x_{N+1}) = \lim_{\sigma \to \infty} K^f_{N+1,N+1} - K^f_{N+1,1:N} \left( K^f_{1:N,1:N} + \sigma^2 K^w_{1:N,1:N} \right)^{-1} K^f_{1:N,N+1}$$

$$= K^f_{N+1,N+1}.$$

Thus, the relaxed solution converges to the optimal solution for $\sigma \to \infty$, i.e.,

$$\lim_{\sigma \to \infty} \overline{f}^\sigma(x_{N+1}) = \lim_{\sigma \to \infty} f^\mu_\sigma + \beta_\sigma \sqrt{\Sigma^f_\sigma(x_{N+1})}$$

$$= \sqrt{K^f_{N+1,N+1}} \Gamma_f$$

$$= \overline{f}_1(x_{N+1}).$$

## C.2  Case 2: Function constraint inactive

We proceed by considering the case in which only (C.1b) is active, proving the result given in Proposition 2.

**Optimal solution**  Problem (C.1) under this active set is given as

$$\overline{f}_2(x_{N+1}) = \sup_{\theta \in \mathbb{R}^r} \quad \Phi_{N+1}\theta \tag{C.4a}$$

$$\text{s.t.} \quad (y - \Phi_{1:N}\theta)^\top (K^w_{1:N,1:N})^{-1} (y - \Phi_{1:N}\theta) - \Gamma^2_w \le 0. \tag{C.4b}$$

This optimization problem only has a finite optimal cost if the span of $\Phi^\top_{N+1} \in \mathbb{R}^{r \times 1}$ is contained in the span of $\Phi^\top_{1:N} \in \mathbb{R}^{r \times N}$, i.e., if $\mathrm{span}(\Phi^\top_{N+1}) \subseteq \mathrm{span}(\Phi^\top_{1:N})$. Otherwise, there would exist a direction $d_1 \in \mathrm{span}(\Phi^\top_{N+1})$ such that $\Phi_{1:N}d_1 = 0$: the optimal solution to (C.4) would then be unbounded and thus would not satisfy the constraint (C.1b) of the original problem. Hence, in the following, we focus on the case where $\mathrm{span}(\Phi^\top_{N+1}) \subseteq \mathrm{span}(\Phi^\top_{1:N})$.

If $\mathrm{span}(\Phi^\top_{N+1}) \subseteq \mathrm{span}(\Phi^\top_{1:N})$, we can write $\Phi^\top_{N+1}$ as a linear combination of the column vectors of $\Phi^\top_{1:N}$, i.e., $\Phi^\top_{N+1} = \Phi^\top_{1:N}\lambda$, where $\lambda \in \mathbb{R}^{N \times 1}$. Since the $r$ feature vectors in

$$\Phi_{1:N+1} = \begin{bmatrix} \Phi_{1:N} \\ \Phi_{N+1} \end{bmatrix} = \begin{bmatrix} I_N \\ \lambda^\top \end{bmatrix} \Phi_{1:N} \tag{C.5}$$

are linearly independent, $\Phi_{1:N}$ has full column rank. As $\Phi^\top_{1:N}$ thus has full row rank, $\lambda$ can be determined as $\lambda = \Phi_{1:N}(\Phi^\top_{1:N}\Phi_{1:N})^{-1}\Phi^\top_{N+1}$.

To reformulate constraint (C.4b) as a norm-ball constraint, we employ a QR decomposition. Recalling the upper-triangular Cholesky factor $R^w_{1:N,1:N}(R^w_{1:N,1:N})^\top = K^w_{1:N,1:N}$ of the noise covariance

matrix from (A.11), we factor the matrix

$$(R^w_{1:N,1:N})^{-1}\Phi_{1:N} = \underbrace{\begin{bmatrix} \tilde{Q}_{11} & \tilde{Q}_{12} \\ \tilde{Q}_{21} & \tilde{Q}_{22} \end{bmatrix}}_{\doteq \tilde{Q}} \begin{bmatrix} \tilde{R} \\ 0 \end{bmatrix}, \tag{C.6}$$

to obtain an orthonormal matrix $\tilde{Q} \in \mathbb{R}^{N \times N}$, with $\tilde{Q}^\top \tilde{Q} = I$, and an upper-triangular matrix $\tilde{R} \in \mathbb{R}^{r \times r}$. The QR factorization implies the following relations required for the proof:

$$\tilde{Q}\tilde{Q}^\top = I \tag{C.7a}$$

$$\begin{bmatrix} \tilde{Q}_{11} \\ \tilde{Q}_{21} \end{bmatrix} = (R^w_{1:N,1:N})^{-1}\Phi_{1:N}\tilde{R}^{-1}, \tag{C.7b}$$

$$\tilde{R}^\top \tilde{R} = \tilde{R}^\top \tilde{Q}^\top \tilde{Q}\tilde{R} = \Phi_{1:N}^\top (K^w_{1:N,1:N})^{-1}\Phi_{1:N}, \tag{C.7c}$$

$$\begin{bmatrix} \tilde{Q}_{12} \\ \tilde{Q}_{22} \end{bmatrix} \begin{bmatrix} \tilde{Q}_{12}^\top & \tilde{Q}_{22}^\top \end{bmatrix} = I - \begin{bmatrix} \tilde{Q}_{11} \\ \tilde{Q}_{21} \end{bmatrix} \begin{bmatrix} \tilde{Q}_{11}^\top & \tilde{Q}_{21}^\top \end{bmatrix}. \tag{C.7d}$$

This allows to write the constraint (C.4b) as

$$\Gamma_w^2 \geq \left\| (R^w_{1:N,1:N})^{-1}(y - \Phi_{1:N}\theta) \right\|_2^2$$

$$\overset{(C.6)}{=} \left\| (R^w_{1:N,1:N})^{-1}y - \tilde{Q}\begin{bmatrix} \tilde{R} \\ 0 \end{bmatrix}\theta \right\|_2^2$$

$$\overset{(C.7a)}{=} \left\| \tilde{Q}^\top (R^w_{1:N,1:N})^{-1}y - \begin{bmatrix} \tilde{R} \\ 0 \end{bmatrix}\theta \right\|_2^2$$

$$\overset{(C.7d)}{=} \left\| \begin{bmatrix} \tilde{Q}_{11}^\top & \tilde{Q}_{21}^\top \end{bmatrix}(R^w_{1:N,1:N})^{-1}y - \tilde{R}\theta \right\|_2^2$$

$$+ \left\| (R^w_{1:N,1:N})^{-1}y \right\|_2^2 - \left\| \begin{bmatrix} \tilde{Q}_{11}^\top & \tilde{Q}_{21}^\top \end{bmatrix}(R^w_{1:N,1:N})^{-1}y \right\|_2^2$$

$$\overset{(C.7b)}{=} \left\| \tilde{R}^{-\top}\Phi_{1:N}^\top (K^w_{1:N,1:N})^{-1}y - \tilde{R}\theta \right\|_2^2$$

$$+ \left\| (R^w_{1:N,1:N})^{-1}y \right\|_2^2 - \left\| \tilde{R}^{-\top}\Phi_{1:N}^\top (K^w_{1:N,1:N})^{-1}y \right\|_2^2$$

$$\overset{(C.7c)}{=} \|z_1\|_2^2 + \tilde{y}^\top \left( K^w_{1:N,1:N} - M \right)\tilde{y},$$

where we used the following definitions in the last line:

$$z_1 \doteq \tilde{R}^{-\top}\Phi_{1:N}^\top (K^w_{1:N,1:N})^{-1}y - \tilde{R}\theta, \tag{C.8}$$

$$\tilde{y} \doteq (K^w_{1:N,1:N})^{-1}y, \tag{C.9}$$

$$M \doteq \Phi_{1:N}\left( \Phi_{1:N}^\top (K^w_{1:N,1:N})^{-1}\Phi_{1:N} \right)^{-1}\Phi_{1:N}^\top. \tag{C.10}$$

Using the coordinate transformation (C.8) and $\Phi_{N+1}^\top = \Phi_{1:N}^\top \lambda$, the cost (C.4a) is rewritten as

$$\lambda^\top \Phi_{1:N}\theta = \lambda^\top \Phi_{1:N}\tilde{R}^{-1}\left( \tilde{R}^{-\top}\Phi_{1:N}^\top (K^w_{1:N,1:N})^{-1}y - z_1 \right)$$

$$= \lambda^\top M\tilde{y} - \lambda^\top \Phi_{1:N}\tilde{R}^{-1}z_1, \tag{C.11}$$

leading to the formulation of (C.4) in the transformed coordinates:

$$\overline{f}_2(x_{N+1}) = \sup_{z_1 \in \mathbb{R}^r} \quad \lambda^\top M\tilde{y} - \lambda^\top \Phi_{1:N}\tilde{R}^{-1}z_1 \tag{C.12a}$$

$$\text{s.t.} \quad \|z_1\|_2^2 \leq \left( \Gamma_w^2 - \tilde{y}^\top \left( K^w_{1:N,1:N} - M \right)\tilde{y} \right). \tag{C.12b}$$

Noting that, by Assumption 1, the right-hand side of the constraint (C.12b) is non-negative, the optimal solution of the above problem is given as

$$z_1^\star = -\frac{\tilde{R}^{-\top}\Phi_{1:N}^\top \lambda}{\|\tilde{R}^{-\top}\Phi_{1:N}^\top \lambda\|_2}\sqrt{\Gamma_w^2 - \tilde{y}^\top \left( K^w_{1:N,1:N} - M \right)\tilde{y}}, \tag{C.13}$$

leading to the corresponding optimal cost (13):

$$\overline{f}_2(x_{N+1}) = \sqrt{\lambda^\top M \lambda} \sqrt{\Gamma_w^2 - \tilde{y}^\top \left(K_{1:N,1:N}^w - M\right) \tilde{y}} + \lambda^\top M \tilde{y}, \tag{C.14}$$

$$\stackrel{\text{(C.7c),(C.10)}}{=} \|P\Phi_{N+1}^\top\|_{P^{-1}} \sqrt{\Gamma_w^2 - y^\top \left(K_{1:N,1:N}^w\right)^{-1} y + \|\theta^\mu\|_{P^{-1}}^2} + \Phi_{N+1}\theta^\mu, \tag{C.15}$$

where (C.15) follows by utilizing that $\Phi_{1:N}^\top \lambda = \Phi_{N+1}^\top$ as well as by defining the weighting matrix

$$P \doteq \left(\Phi_{1:N}^\top \left(K_{1:N,1:N}^w\right)^{-1} \Phi_{1:N}\right)^{-1}$$

and the least-squares estimator for the unknown parameters

$$\theta^\mu \doteq P\Phi_{1:N}^\top \left(K_{1:N,1:N}^w\right)^{-1} y.$$

**Feasibility check** In the original coordinates, the optimal solution is given as

$$\theta^{\star,2} = \tilde{R}^{-1}\tilde{R}^{-\top}\Phi_{1:N}^\top(K_{1:N,1:N}^w)^{-1}y - \tilde{R}^{-1}z_1^\star$$

$$\stackrel{\text{(C.7c)}}{=} \left(\Phi_{1:N}^\top(K_{1:N,1:N}^w)^{-1}\Phi_{1:N}\right)^{-1}\Phi_{1:N}^\top \left(\tilde{y} + \lambda \frac{\sqrt{\Gamma_w^2 - \tilde{y}^\top \left(K_{1:N,1:N}^w - M\right)\tilde{y}}}{\sqrt{\lambda^\top M \lambda}}\right)$$

$$= \theta^\mu + \frac{P\Phi_{N+1}^\top}{\|P\Phi_{N+1}^\top\|_{P^{-1}}}\sqrt{\Gamma_w^2 - y^\top \left(K_{1:N,1:N}^w\right)^{-1} y + \|\theta^\mu\|_{P^{-1}}^2}; \tag{C.16}$$

the point $\theta^{\star,2}$ is feasible for the original problem (C.1) if it satisfies the neglected constraint (C.1b), i.e., if $\|\theta^{\star,2}\|_2^2 \leq \Gamma_f^2$, retrieving (12).

**Connection to relaxed solution** The quantities in the optimal cost (C.14) can be expressed as limiting values related to the relaxed solution for $\sigma \to 0$. For the matrix $M$ in (C.10), it holds that

$$M = \Phi_{1:N}\left(\Phi_{1:N}^\top(K_{1:N,1:N}^w)^{-1}\Phi_{1:N}\right)^{-1}\Phi_{1:N}^\top$$

$$= \lim_{\sigma \to 0} \Phi_{1:N}\left(\sigma^2 I_r + \Phi_{1:N}^\top(K_{1:N,1:N}^w)^{-1}\Phi_{1:N}\right)^{-1}\Phi_{1:N}^\top$$

$$= \lim_{\sigma \to 0} \frac{1}{\sigma^2}\Phi_{1:N}\left(I_r - \Phi_{1:N}^\top\left((\sigma^2 K_{1:N,1:N}^w) + \Phi_{1:N}\Phi_{1:N}^\top\right)^{-1}\Phi_{1:N}\right)\Phi_{1:N}^\top$$

$$= \lim_{\sigma \to 0} \frac{1}{\sigma^2}\left(\Phi_{1:N}\Phi_{1:N}^\top - \Phi_{1:N}\Phi_{1:N}^\top\left((\sigma^2 K_{1:N,1:N}^w) + \Phi_{1:N}\Phi_{1:N}^\top\right)^{-1}\Phi_{1:N}\Phi_{1:N}^\top\right).$$

With $\lambda^\top \Phi_{1:N} = \Phi_{N+1}$, this results in

$$\lambda^\top M \lambda = \lim_{\sigma \to 0} \frac{1}{\sigma^2}\left(\Phi_{N+1}\Phi_{N+1}^\top - \Phi_{N+1}\Phi_{1:N}^\top\left((\sigma^2 K_{1:N,1:N}^w) + \Phi_{1:N}\Phi_{1:N}^\top\right)^{-1}\Phi_{1:N}\Phi_{N+1}^\top\right)$$

$$= \lim_{\sigma \to 0} \frac{1}{\sigma^2}\left(K_{N+1,N+1}^f - K_{N+1,1:N}^f\left((\sigma^2 K_{1:N,1:N}^w) + K_{1:N,1:N}^f\right)^{-1}K_{1:N,N+1}^f\right)$$

$$\stackrel{\text{(5b)}}{=} \lim_{\sigma \to 0} \frac{1}{\sigma^2}\Sigma_\sigma^f(x_{N+1}). \tag{C.17}$$

The offset $\lambda^\top M \tilde{y}$ in (C.14) is equivalent to the offset $f_\sigma^\mu(x_{N+1})$ in the optimal cost (C.1a) for the relaxed problem:

$$\lambda^\top M \tilde{y} = \lim_{\sigma \to 0} \frac{1}{\sigma^2}\left(\Phi_{N+1}\Phi_{1:N}^\top - \Phi_{N+1}\Phi_{1:N}^\top\left((\sigma^2 K_{1:N,1:N}^w) + \Phi_{1:N}\Phi_{1:N}^\top\right)^{-1}\Phi_{1:N}\Phi_{1:N}^\top\right)(K_{1:N,1:N}^w)^{-1}y$$

$$= \lim_{\sigma \to 0} \Phi_{N+1}\Phi_{1:N}^\top\left((\sigma^2 K_{1:N,1:N}^w) + \Phi_{1:N}\Phi_{1:N}^\top\right)^{-1}\left(\Phi_{1:N}\Phi_{1:N}^\top - \Phi_{1:N}\Phi_{1:N}^\top + \sigma^2 K_{1:N,1:N}^w\right)(\sigma^2 K_{1:N,1:N}^w)^{-1}y$$

$$= \lim_{\sigma \to 0} K_{N+1,1:N}^f\left((\sigma^2 K_{1:N,1:N}^w) + K_{1:N,1:N}^f\right)^{-1}y$$

$$\overset{(5a)}{=} \lim_{\sigma \to 0} f^\mu_\sigma(x_{N+1}) \tag{C.18}$$

For the last term, $\Gamma^2_w - \tilde{y}^\top \left( K^w_{1:N,1:N} - M \right) \tilde{y}$, we have that

$$\Gamma^2_w - \tilde{y}^\top \left( K^w_{1:N,1:N} - M \right) \tilde{y} = \lim_{\sigma \to 0} \sigma^2 \Gamma^2_f + \Gamma^2_w - \tilde{y}^\top \left( K^w_{1:N,1:N} - M \right) \tilde{y}$$

$$= \lim_{\sigma \to 0} \sigma^2 \left( \Gamma^2_f + \frac{\Gamma^2_w}{\sigma^2} - \frac{1}{\sigma^2} \tilde{y}^\top \left( K^w_{1:N,1:N} - M \right) \tilde{y} \right)$$

$$= \lim_{\sigma \to 0} \sigma^2 \left( \Gamma^2_f + \frac{\Gamma^2_w}{\sigma^2} - y^\top \left( W^{-1} - W^{-1}(\sigma^2 M)W^{-1} \right) y \right), \tag{C.19}$$

where we have introduced the abbreviation $W \doteq \sigma^2 K^w_{1:N,1:N}$ for notational simplicity; similarly, we abbreviate $F \doteq K^f_{1:N,1:N}$. Recalling that $M = \frac{1}{\sigma^2}(F - F(W + F)^{-1}F)$ and by using [Searle and Khuri, 2017, Exercise 16.(d), Chapter 5], the data-dependent term in the above expression can be simplified as follows:

$$W^{-1} - W^{-1}\left( F - F\left( W + F \right)^{-1} F \right) W^{-1}$$

$$= W^{-1} - W^{-1}FW^{-1} + W^{-1}FW^{-1} \left( W^{-1} + W^{-1}FW^{-1} \right)^{-1} W^{-1}FW^{-1}$$

$$= W^{-1} - (W^{-1} - W^{-1}(W^{-1}FW^{-1} + W^{-1})^{-1}W^{-1})$$

$$= W^{-1}(W^{-1} + W^{-1}FW^{-1})^{-1}W^{-1} = (W + F)^{-1}.$$

To summarize, it holds that

$$\Gamma^2_w - \tilde{y}^\top \left( K^w_{1:N,1:N} - M \right) \tilde{y} = \lim_{\sigma \to 0} \sigma^2 \Gamma^2_f + \Gamma^2_w - \tilde{y}^\top \left( K^w_{1:N,1:N} - M \right) \tilde{y}$$

$$= \lim_{\sigma \to 0} \sigma^2 \left( \Gamma^2_f + \frac{\Gamma^2_w}{\sigma^2} - y^\top \left( K^f_{1:N,1:N} + (\sigma^2 K^w_{1:N,1:N}) \right)^{-1} y \right)$$

$$= \lim_{\sigma \to 0} \sigma^2 \left( \Gamma^2_f + \frac{\Gamma^2_w}{\sigma^2} - \|g^\mu_\sigma\|^2_{\mathcal{H}_{k^f + k^\sigma}} \right)$$

and the total bound for Case 2 is given as

$$\overline{f}_2(x_{N+1}) = \sqrt{\lambda^\top M \lambda} \sqrt{\Gamma^2_w - \tilde{y}^\top \left( K^w_{1:N,1:N} - M \right) \tilde{y}} + \lambda^\top M \tilde{y}$$

$$= \sqrt{\lim_{\sigma \to 0} \frac{1}{\sigma^2} \Sigma^f_\sigma(x_{N+1})} \sqrt{\lim_{\sigma \to 0} \sigma^2 \left( \Gamma^2_f + \frac{\Gamma^2_w}{\sigma^2} - \|g^\mu_\sigma\|^2_{\mathcal{H}_{k^f + k^\sigma}} \right)} + \lim_{\sigma \to 0} f^\mu_\sigma$$

$$= \lim_{\sigma \to 0} \left( \sqrt{\Sigma^f_\sigma(x_{N+1})} \sqrt{\Gamma^2_f + \frac{\Gamma^2_w}{\sigma^2} - \|g^\mu_\sigma\|^2_{\mathcal{H}_{k^f + k^\sigma}}} + f^\mu_\sigma \right)$$

$$= \lim_{\sigma \to 0} \overline{f}^\sigma(x_{N+1}).$$

**Proof of Corollary 1**

We now prove Corollary 1, which simplifies the general result of Proposition 2 under the assumptions that the kernel matrix $K^f_{1:N,1:N}$ is invertible and the test point is equal to the $k$-th training point, i.e., $x_{N+1} = x_k$, for some $k \in \{1, \dots, N\}$. In this case, the $k$-th and $(N+1)$-th row of the kernel matrix $K^f_{1:N+1,1:N+1}$ are identical. In terms of the singular value decomposition, by (A.5) this implies that

$$K^f_{k,1:N} = e^\top_k \Phi_{1:N} \Phi^\top_{1:N} = \Phi_{N+1} \Phi^\top_{1:N} = K^f_{N+1,1:N},$$

i.e., the relation $\Phi^\top_{N+1} = \Phi^\top_{1:N} \lambda$ holds for $\lambda = e_k$, with $e_k$ being the $k$-th unit vector. Since $\Phi_{1:N} \in \mathbb{R}^{r \times N}$ has rank $r = N$, it is invertible. This allows to simplify the expression for the optimal cost using that $M\tilde{y} = y$, with $M$ and $\tilde{y}$ defined as in Eqs. (C.9) and (C.10):

$$\overline{f}_2(x_k) = \sqrt{e^\top_k M e_k} \sqrt{\Gamma^2_w} + e^\top_k y$$

$$= \sqrt{e_k^\top \Phi_{1:N} \left( \Phi_{1:N}^\top (K_{1:N,1:N}^w)^{-1} \Phi_{1:N} \right)^{-1} \Phi_{1:N}^\top e_k \Gamma_w + e_k^\top y}$$

$$= \sqrt{e_k^\top K_{1:N,1:N}^w e_k \Gamma_w + e_k^\top y}$$

$$= \sqrt{K_{k,k}^w \Gamma_w + y_k},$$

which is the optimal cost as given in (15). By inserting the simplified expressions into (C.16), the optimal $\theta$ becomes

$$\theta^{\star,2} = \left( \Phi_{1:N}^\top (K_{1:N,1:N}^w)^{-1} \Phi_{1:N} \right)^{-1} \Phi_{1:N}^\top \left( \tilde{y} + e_k \frac{\sqrt{\Gamma_w^2}}{\sqrt{e_k^\top M e_k}} \right)$$

$$= \Phi_{1:N}^{-1} \left( y + K_{1:N,k}^w \frac{\Gamma_w}{\sqrt{K_{k,k}^w}} \right).$$

Recalling the low-rank factorization of $K_{1:N,1:N}^f$ in (A.7), the feasibility condition based on the neglected constraint (C.1b) reduces to

$$\|\theta^{\star,2}\|_2^2 = \left\| \left( y + K_{1:N,k}^w \frac{\Gamma_w}{\sqrt{K_{k,k}^w}} \right) \right\|_{(K_{1:N,1:N}^f)^{-1}}^2 \leq \Gamma_f^2,$$

as presented in (14).

*Remark.* Corollary 1 has been derived as a particular case of Proposition 2, which provides the analytic solution for the case $\sigma \to 0$ occurring when the kernel matrix is rank-deficient. From this perspective, the scenario in which the test-input belongs to the training data-set is one of the particular situations leading to a drop in the rank of the kernel matrix. However, the proof of Corollary 1 could be alternatively carried out following the steps of the one of Proposition 1.

### C.3   Case 3: Both constraints active

Next, we consider the case when both constraints (C.1b) and (C.1c) are active.

**Optimal solution**   We first show that strong duality holds for both the relaxed problem (A.3) as well as the original problem (C.1). Afterwards, we establish that there exists a value $\sigma \in (0, \infty)$, such that the primal optimizer of the relaxed problem is a primal optimizer for the original problem.

For the original problem (C.1), we show that a strictly feasible solution can be constructed using the true latent function and noise process. Let $f^{\mathrm{tr,int}} \in \mathcal{H}_{k^f}$ be the minimum-norm interpolant of the latent function at the test and training input locations, i.e.,

$$f^{\mathrm{tr,int}} = \underset{f \in \mathcal{H}_{k^f}}{\arg\min} \quad \|f\|_{\mathcal{H}_{k^f}}^2 \tag{C.20a}$$

$$\text{s.t.} \quad f(x_i) = f^{\mathrm{tr}}(x_i), \; i = 1, \dots, N+1. \tag{C.20b}$$

Similarly, let $w^{\mathrm{tr,int}} \in \mathcal{H}_{k^w}$ be the minimum-norm interpolant of the noise-generating process at the training input locations (excluding the test point),

$$w^{\mathrm{tr,int}} = \underset{w \in \mathcal{H}_{k^w}}{\arg\min} \quad \|w\|_{\mathcal{H}_{k^w}}^2 \tag{C.21a}$$

$$\text{s.t.} \quad w(x_i) = w^{\mathrm{tr}}(x_i), \; i = 1, \dots, N. \tag{C.21b}$$

The representer theorem [Kimeldorf and Wahba, 1971] establishes that the solutions to the above optimization problems is finite-dimensional and given by

$$f^{\mathrm{tr,int}}(\cdot) = \sum_{i=1}^{N+1} k^f(\cdot, x_i)\alpha_i^{f,\mathrm{tr}}, \quad w^{\mathrm{tr,int}}(\cdot) = \sum_{i=1}^{N} k^w(\cdot, x_i)\alpha_i^{w,\mathrm{tr}}.$$

By design, the sum of both functions interpolates the training data, i.e., $f^{\mathrm{tr,int}}(x_i) + w^{\mathrm{tr,int}}(x_i) = y_i$ for $i = 1, \ldots, N$. Additionally, by Assumption 1, it holds that $f^{\mathrm{tr,int}}$ and $w^{\mathrm{tr,int}}$ satisfy their corresponding RKHS-norm bound, i.e., $\|f^{\mathrm{tr,int}}\|_{\mathcal{H}_{k_f}}^2 \leq \|f^{\mathrm{tr}}\|_{\mathcal{H}_{k_f}}^2 < \Gamma_f^2$ and $\|w^{\mathrm{tr,int}}\|_{\mathcal{H}_{k_w}}^2 \leq \|w^{\mathrm{tr}}\|_{\mathcal{H}_{k_w}}^2 < \Gamma_w^2$. Thus, the corresponding coefficient vector $\theta^{\mathrm{tr}} \stackrel{(A.7)}{=} S_r^{1/2} v_1^{\mathrm{tr}} = S_r^{1/2} \begin{bmatrix} V_{11}^\top & V_{12}^\top \end{bmatrix} \alpha^{f,\mathrm{tr}}$ constitutes a strictly feasible solution of the finite-dimensional problem formulation (C.1). This implies that Slater's condition is satisfied for the convex program (C.1), which implies that strong duality holds. Since every strictly feasible solution for the original problem (C.1) is also strictly feasible for the relaxed problem (B.4), similarly, Slater's condition and strong duality hold for the relaxed problem.

Due to strong duality, the point $\theta^{\star,\sigma}$ is the unique minimizer of the relaxed problem (A.3) if and only if the primal-dual pair $(\theta^{\star,\sigma}, \lambda_g^{\star,\sigma})$ satisfies the KKT conditions

$$\nabla_\theta \mathcal{L}_\sigma(\theta^{\star,\sigma}, \lambda_g^{\star,\sigma}) = 0, \qquad \text{(C.22a)}$$

$$\left(\|\theta^{\star,\sigma}\|_2^2 - \Gamma_f^2\right) + \frac{1}{\sigma^2}\left(\|y - \Phi_{1:N}\theta^{\star,\sigma}\|_{(K_{1:N,1:N}^w)^{-1}}^2 - \Gamma_w^2\right) \leq 0, \qquad \text{(C.22b)}$$

$$\lambda_g^{\star,\sigma}\left(\left(\|\theta^{\star,\sigma}\|_2^2 - \Gamma_f^2\right) + \frac{1}{\sigma^2}\left(\|y - \Phi_{1:N}\theta^{\star,\sigma}\|_{(K_{1:N,1:N}^w)^{-1}}^2 - \Gamma_w^2\right)\right) = 0, \qquad \text{(C.22c)}$$

$$\lambda_g^{\star,\sigma} \geq 0, \qquad \text{(C.22d)}$$

with the corresponding Lagrangian

$$\mathcal{L}_\sigma(\theta, \lambda_g) = \Phi_{N+1}\theta - \lambda_g\left(\|\theta\|_2^2 - \Gamma_f^2 + \frac{1}{\sigma^2}\left(\|y - \Phi_{1:N}\theta\|_{(K_{1:N,1:N}^w)^{-1}}^2 - \Gamma_w^2\right)\right)$$

$$= \Phi_{N+1}\theta - \lambda_g\left(\|\theta\|_2^2 - \Gamma_f^2\right) - \frac{\lambda_g}{\sigma^2}\left(\|y - \Phi_{1:N}\theta\|_{(K_{1:N,1:N}^w)^{-1}}^2 - \Gamma_w^2\right).$$

Similarly, due to strong duality, the point $\theta^\star$ is the unique minimizer if the original problem (C.1) if and only if the primal-dual pair $(\theta^\star, \lambda_f^\star, \lambda_w^{\star,\sigma})$ satisfies the KKT conditions

$$\nabla_\theta \mathcal{L}(\theta^\star, \lambda_f^\star, \lambda_w^{\star,\sigma}) = 0$$

$$\|\theta^\star\|_2^2 - \Gamma_f^2 \leq 0$$

$$\|y - \Phi_{1:N}\theta^\star\|_{(K_{1:N,1:N}^w)^{-1}}^2 - \Gamma_w^2 \leq 0$$

$$\lambda_f^\star\left(\|\theta^\star\|_2^2 - \Gamma_f^2\right) = 0$$

$$\lambda_w^{\star,\sigma}\left(\|y - \Phi_{1:N}\theta^\star\|_{(K_{1:N,1:N}^w)^{-1}}^2 - \Gamma_w^2\right) = 0$$

$$\lambda_f^\star, \lambda_w^{\star,\sigma} \geq 0$$

with corresponding Lagrangian

$$\mathcal{L}(\theta, \lambda_f, \lambda_w) = \Phi_{N+1}\theta - \lambda_f\left(\|\theta\|_2^2 - \Gamma_f^2\right) - \lambda_w\left(\|y - \Phi_{1:N}\theta\|_{(K_{1:N,1:N}^w)^{-1}} - \Gamma_w^2\right).$$

Let $(\theta^{\star,3}, \lambda_f^{\star,3}, \lambda_w^{\star,3})$ be the optimal primal-dual solution satisfying the KKT conditions of the original problem (C.1) under the imposed active set. Since the constraints (C.1b) and (C.1c) are active, it holds that $\lambda_f^{\star,3}, \lambda_w^{\star,3} > 0$. Now, let $(\sigma^\star)^2 = \frac{\lambda_f^{\star,3}}{\lambda_w^{\star,3}}$. Then, $\lambda_f^{\star,3} = (\sigma^\star)^2 \lambda_w^{\star,3}$ and the primal-dual pair $(\theta^{\star,3}, \lambda_f^{\star,3})$ satisfy the KKT conditions (C.22) of the relaxed problem:

1. Since $\nabla_\theta \mathcal{L}_\sigma(\theta^{\star,3}, \lambda_f^{\star,3}) = \nabla_\theta \mathcal{L}(\theta^{\star,3}, \lambda_f^{\star,3}, \lambda_w^{\star,3}) = 0$, the stationarity condition is fulfilled.

2. As both constraints (C.1b) and (C.1c) are active,
$$\|\theta^{\star,3}\|_2^2 - \Gamma_f^2 = -\left(\|y - \Phi_{1:N}\theta^{\star,3}\|_{(K_{1:N,1:N}^w)^{-1}}^2 - \Gamma_w^2\right) = 0,$$
i.e., primal feasibility and complementarity slackness are fulfilled.

3. The optimal multiplier $\lambda_g^{\star,\sigma} = \lambda_f^{\star,3} > 0$ for the relaxed problem is positive.

Hence, due to strong duality since both constraints (C.1b) and (C.1c) are active, $(\theta^{\star,3}, \lambda_f^{\star,3})$ is the optimal primal-dual solution for the relaxed problem (A.3) with $\sigma = \sigma^\star = \sqrt{\lambda_f^{\star,3}/\lambda_w^{\star,3}}$ if and only if $(\theta^{\star,3}, \lambda_f^{\star,3}, \lambda_w^{\star,3})$ is the optimal primal-dual solution for the original problem (C.1).

**Feasibility check**    The solution is feasible by definition.

**Connection to relaxed solution**    As shown above, the optimal cost can be recovered by the cost of the relaxed problem for a specific choice of noise parameter $\sigma = \sigma^\star$, i.e.,

$$\overline{f}_3(x_{N+1}) = \overline{f}^{\sigma^\star}(x_{N+1}). \tag{C.23}$$

## C.4    Case 4: Both constraints inactive

Last, we investigate the case when both constraints (C.1b) and (C.1c) are inactive.

**Optimal solution**    The optimal solution to the unconstrained linear program is given by case distinction:

$$\overline{f}_4(x_{N+1}) = \sup_{\theta \in \mathbb{R}^r} \quad \Phi_{N+1}\theta \tag{C.24}$$

$$= \begin{cases} 0, & \text{if } \Phi_{N+1} = 0, \\ \infty, & \text{if } \Phi_{N+1} \neq 0. \end{cases}$$

**Feasibility check**    For $\Phi_{N+1} \neq 0$, as the optimal solution is unbounded, it is infeasible for the original problem (C.1), whose feasible set is compact, in particular due to constraint (C.1b). Hence, this case never corresponds to the optimal solution.

If $\Phi_{N+1} = 0$, any $\theta^{\star,4} \in \mathbb{R}^r$ is optimal. Thus, any feasible solution satisfying constraints (C.1b) and (C.1c) is also optimal.

**Connection to relaxed solution**    If $\Phi_{N+1} = 0$, the solution of the relaxed problem (A.3) is given by $\overline{f}^\sigma(x_{N+1}) = 0$, i.e., it recovers the solution of the original problem (C.24) for any $\sigma \in (0, \infty)$.

## C.5    Finding the correct active set

Let $\theta^{\star,i}$ denote the primal solution corresponding to Case $i$, with $i = 1, \ldots, 4$. The active set for the optimal solution is given by the one for which the corresponding primal solution $\theta^{\star,i}$ is feasible for the original problem and leads to the maximum cost among all feasible optimizers $\theta^{\star,i}$ for a specific active set, i.e.,

$$\overline{f}(x_{N+1}) = \max_{i \in \{1,2,3,4\}} \quad \Phi_{N+1}\theta^{\star,i} \tag{C.25a}$$

$$\text{s.t.} \quad \|\theta^{\star,i}\|_2^2 \leq \Gamma_f^2, \tag{C.25b}$$

$$\|y - \Phi_{1:N}\theta^{\star,i}\|_{(K_{1:N,1:N}^w)^{-1}}^2 \leq \Gamma_w^2. \tag{C.25c}$$

The solution to the above problem can be obtained by case distinction. The optimal cost for a subset of active constraints lower-bounds the optimal cost for a superset, i.e., $\Phi_{N+1}\theta^{\star,4} \geq \Phi_{N+1}\theta^{\star,j} \geq \Phi_{N+1}\theta^{\star,3}$ for $j \in \{1,2\}$. Thus, if $\theta^{\star,4}$ is feasible, it will be optimal. Otherwise, if either $\theta^{\star,1}$ or $\theta^{\star,2}$ is feasible, it will be optimal. If neither of the other cases is feasible, $\theta^{\star,3}$ is the optimal solution.

Now, we compare the optimal cost if both $\theta^{\star,1}$ and $\theta^{\star,2}$ are feasible. If $\theta^{\star,1}$ is a feasible solution of (C.1), then it holds that the neglected constraint (C.1c) does not change the optimal solution of (C.1). Similarly, if $\theta^{\star,1}$ is a feasible solution of (C.1), then it holds that the neglected constraint (C.1b) does not change the optimal solution of (C.1). Combining both facts, it holds that

$$\Phi_{N+1}\theta^{\star,1} = \Phi_{N+1}\theta^\star = \Phi_{N+1}\theta^{\star,2},$$

i.e., the optimal cost in Case 1 and Case 2 is equal, $\overline{f}_1(x_{N+1}) = \overline{f}_2(x_{N+1})$. Finally, we note that in Case 4 it holds that $\Phi_{N+1}\theta^\star = \Phi_{N+1}\theta^{\star,4} = 0 = \Phi_{N+1}\theta^{\star,j}$, $j \in \{1,2,3\}$, i.e., the optimal cost in all four cases is equal. Therefore, it does not need to be considered explicitly.

To summarize, the optimal cost is determined as follows:

$$
\overline{f}(x_{N+1}) = \begin{cases} \sqrt{K^f_{N+1,N+1}}\Gamma_f, & \text{in Cases 1 and 4,} \\ \Phi_{N+1}\theta^\mu + \|P\Phi^\top_{N+1}\|_{P^{-1}}\sqrt{\Gamma_w^2 - y^\top\left(K^w_{1:N,1:N}\right)^{-1}y + \|\theta^\mu\|^2_{P^{-1}}} & \text{in Cases 2 and 4,} \\ f^\mu_{\sigma^\star}(x_{N+1}) + \sqrt{\Gamma_f^2 + \frac{\Gamma_w^2}{(\sigma^\star)^2} - \|g^\mu_{\sigma^\star}\|^2_{\mathcal{H}_{k_f + k_{\sigma^\star}}}}\sqrt{\Sigma^f_{\sigma^\star}(x_{N+1})}, & \text{in Cases 3 and 4.} \end{cases}
$$

$$
= \begin{cases} \lim_{\sigma\to\infty}\overline{f}^\sigma(x_{N+1}), & \text{if } \left\|y - K^f_{1:N,N+1}\dfrac{\Gamma_f}{\sqrt{K^f_{N+1,N+1}}}\right\|^2_{(K^w_{1:N,1:N})^{-1}} \leq \Gamma_w^2\,, \\ \lim_{\sigma\to 0}\overline{f}^\sigma(x_{N+1}), & \text{if } \left\|\theta^\mu + \dfrac{P\Phi^\top_{N+1}}{\|P\Phi^\top_{N+1}\|_{P^{-1}}}\sqrt{\Gamma_w^2 - y^\top\left(K^w_{1:N,1:N}\right)^{-1}y + \|\theta^\mu\|^2_{P^{-1}}}\right\|^2_2 \leq \Gamma_f^2, \\ \inf_{\sigma\in(0,\infty)}\overline{f}^\sigma(x_{N+1}), & \text{otherwise.} \end{cases}
$$

Finally, we show that the analytical solutions in all cases can be reduced to a single expression:

1. Let Case 1 be feasible, i.e., $\overline{f}(x_{N+1}) = \overline{f}_1(x_{N+1}) = \lim_{\sigma\to\infty}\overline{f}^\sigma(x_{N+1})$. Since $\overline{f}_1(x_{N+1}) \leq \overline{f}_3(x_{N+1})$, it holds that $\lim_{\sigma\to\infty}\overline{f}^\sigma(x_{N+1}) \leq \inf_{\sigma\in(0,\infty)}\overline{f}^\sigma(x_{N+1})$. However, for the infimum it also holds that $\inf_{\sigma\in(0,\infty)}\overline{f}^\sigma(x_{N+1}) \leq \lim_{\sigma\to\infty}\overline{f}^\sigma(x_{N+1})$. Therefore, it holds that $\overline{f}(x_{N+1}) = \lim_{\sigma\to\infty}\overline{f}^\sigma(x_{N+1}) = \inf_{\sigma\in(0,\infty)}\overline{f}^\sigma(x_{N+1})$.

2. Let Case 2 be feasible, i.e., $\overline{f}(x_{N+1}) = \overline{f}_2(x_{N+1}) = \lim_{\sigma\to 0}\overline{f}^\sigma(x_{N+1})$. Analogously as above, since $\overline{f}_2(x_{N+1}) \leq \overline{f}_3(x_{N+1})$ and $\inf_{\sigma\in(0,\infty)}\overline{f}^\sigma(x_{N+1}) \leq \lim_{\sigma\to 0}\overline{f}^\sigma(x_{N+1})$, it holds that $\overline{f}(x_{N+1}) = \lim_{\sigma\to 0}\overline{f}^\sigma(x_{N+1}) = \inf_{\sigma\in(0,\infty)}\overline{f}^\sigma(x_{N+1})$.

3. In Case 3, since it holds that $\overline{f}_3(x_{N+1}) = \overline{f}^\sigma(x_{N+1})$ for a specific value of $\sigma = \sigma^\star \in (0,\infty)$, this implies that $\inf_{\sigma\in(0,\infty)}\overline{f}^\sigma(x_{N+1}) \leq \overline{f}_3(x_{N+1})$. However, since any feasible solution $\theta \in \mathbb{R}^r$ of the original problem (C.1) is also feasible for the relaxed problem (A.3) for any $\sigma \in (0,\infty)$, the cost of the original problem is upper-bounded by the cost of the relaxed problem, i.e., it also holds that $\overline{f}_3(x_{N+1}) \leq \inf_{\sigma\in(0,\infty)}\overline{f}^\sigma(x_{N+1})$. Combining both inequalities, it thus follows that $\overline{f}_3(x_{N+1}) = \inf_{\sigma\in(0,\infty)}\overline{f}^\sigma(x_{N+1})$.

Therefore, we have shown that

$$
\overline{f}(x_{N+1}) = \inf_{\sigma\in(0,\infty)}\overline{f}^\sigma(x_{N+1}),
$$

as claimed in (9).

### C.6   Lower bound

The lower bound corresponding to Theorem 1 is obtained by the same steps as the upper bound, replacing "sup" with "inf". In Cases 1 and 2, this leads to a flipped sign for the solutions of the free components in Eqs. (C.3) and (C.13), affecting the feasibility checks. Overall, the optimal lower bound is also obtained by case distinction:

$$
\underline{f}(x_{N+1}) = \begin{cases} -\sqrt{K^f_{N+1,N+1}}\Gamma_f, & \text{in Cases 1 and 4,} \\ \Phi_{N+1}\theta^\mu - \|P\Phi^\top_{N+1}\|_{P^{-1}}\sqrt{\Gamma_w^2 - y^\top\left(K^w_{1:N,1:N}\right)^{-1}y + \|\theta^\mu\|^2_{P^{-1}}} & \text{in Cases 2 and 4,} \\ f^\mu_{\sigma^\star}(x_{N+1}) - \sqrt{\Gamma_f^2 + \frac{\Gamma_w^2}{(\sigma^\star)^2} - \|g^\mu_{\sigma^\star}\|^2_{\mathcal{H}_{k_f + k_{\sigma^\star}}}}\sqrt{\Sigma^f_{\sigma^\star}(x_{N+1})}, & \text{in Cases 3 and 4.} \end{cases}
$$

$$
= \begin{cases} \lim_{\sigma\to\infty}\underline{f}^\sigma(x_{N+1}), & \text{if } \left\|y + K^f_{1:N,N+1}\dfrac{\Gamma_f}{\sqrt{K^f_{N+1,N+1}}}\right\|^2_{(K^w_{1:N,1:N})^{-1}} \leq \Gamma_w^2\,, \\ \lim_{\sigma\to 0}\underline{f}^\sigma(x_{N+1}), & \text{if } \left\|\theta^\mu - \dfrac{P\Phi^\top_{N+1}}{\|P\Phi^\top_{N+1}\|_{P^{-1}}}\sqrt{\Gamma_w^2 - y^\top\left(K^w_{1:N,1:N}\right)^{-1}y + \|\theta^\mu\|^2_{P^{-1}}}\right\|^2_2 \leq \Gamma_f^2, \\ \sup_{\sigma\in(0,\infty)}\underline{f}^\sigma(x_{N+1}), & \text{otherwise.} \end{cases}
$$

Analogous to Appendix C.5, by replacing "inf" with "sup" and flipping the corresponding inequalities, the optimal solution is shown to be given as

$$\underline{f}(x_{N+1}) = \sup_{\sigma \in (0,\infty)} \underline{f^{\sigma}}(x_{N+1}). \tag{C.26}$$

Note that this bound is *not symmetric*, as the supremum and infimum can be attained for different values of the noise parameter $\sigma$.

## D Numerical example on safe control for uncertain nonlinear systems: Implementation details

In the following, we describe the setup of the optimization problem solved in the numerical example "Safe control for uncertain nonlinear systems" in Section 4.3. Inserting the system dynamics (18) into the safety constraint $x(k+1) \geq (1-\gamma)x(k)$, the condition $f^{\mathrm{known}}(x(k), u(k)) + f^{\mathrm{tr}}(x(k), u(k)) \geq (1-\gamma)x(k)$ can be enforced robustly by utilizing the lower uncertainty bound $f_\sigma^\mu(x, u) - \beta_\sigma \sqrt{\Sigma_\sigma(x, u)} \leq f^{\mathrm{tr}}(x, u)$, leading to the tightened constraint

$$(1-\gamma)x(k) \leq f^{\mathrm{known}}(x(k), u(k)) + f_\sigma^\mu(x(k), u(k)) - \beta_\sigma \sqrt{\Sigma_\sigma(x(k), u(k))}.$$

Minimization of the user-defined cost $c(x(k), u(k))$ subject to the above constraint can be achieved by solving the following optimization problem:

$$\min_{x_0, x_1, u_0, \sigma, s} \quad c(x_0, u_0) + \omega s$$
$$\text{s.t.} \quad x_0 = x(k),$$
$$x_1 = f^{\mathrm{known}}(x_0, u_0) + f_\sigma^\mu(x_0, u_0),$$
$$x_1 \geq (1-\gamma)x_0 + \beta_\sigma \sqrt{\Sigma_\sigma(x_0, u_0)} - s,$$
$$s \geq 0,$$
$$u_{\max} \geq u_0 \geq u_{\min}.$$

Therein, the added slack variable $s$ allows the optimizer to converge even in case the decrease condition is impossible to satisfy given the bounds $u_{\min}, u_{\max}$ on the control input. A large linear penalty $\omega > 0$ thereby incentivizes $s = 0$, i.e., constraint satisfaction; a solution is classified as feasible if the optimal slack value $s^\star$ satisfies $s^\star \leq 10^{-6}$. The optimization problems are implemented in `CasADi` [Andersson et al., 2019] and solved using the interior-point optimizer `IPOPT` [Wächter and Biegler, 2006] on an Intel i9-7940X CPU. Table 1 provides all parameters and expressions used for the implementation.

| Parameter | Value |
|---|---|
| $f^{\mathrm{known}}(x, u)$ | $0.5x + u - 1$ |
| $c(x, u)$ | $(f^{\mathrm{known}}(x, u) + f_\sigma^\mu(x, u))^2 + u^2$ |
| $u_{\min}$ | $-2$ |
| $u_{\max}$ | $2$ |
| $\gamma$ | $0.95$ |
| $\omega$ | $10^4$ |

Table 1: Parameters and expressions for the CBF example

Additional implementation details can be found in the published source code at `https://gitlab.ethz.ch/ics/bounded-energy-rkhs-bounds` and at `https://doi.org/10.3929/ethz-c-000785083`.

