# OpenReview forum: "Optimal kernel regression bounds under energy-bounded noise"
_NeurIPS.cc/2025/Conference — NeurIPS 2025 poster_

### Official Review · Reviewer_BU3a · 2025-06-16

**Clarity:** 3
**Significance:** 2
**Originality:** 3
**Rating:** 4
**Confidence:** 4

**Summary:**

The paper considers the problem of obtaining uncertainty bounds for the problem of kernel-regression. As opposed to the standard i.i.d. assumption on noise, the authors consider a bounded-energy scenario, where the noise is assumed to have a bounded norm in a certain RKHS space. Under this setting, the authors derive a closed expression for the worst case upper and lower bounds on the mean predictor providing concise descriptions of the worst-case uncertainty. They also provide simplified expressions under specific cases and relate the obtained bounds to classical results in kernel and linear regression.

**Questions:**

One thing that is bugging me is the premise and the motivation of this problem. At a basic level, the problem of function estimation/prediction using training data first requires a certain modelling assumption. Existing studies on kernel-based regression model the data assuming that the true function belongs to an RKHS and the noise is i.i.d. Gaussian. In this work, the authors use a different modelling assumption, where in they assume that noise is an element of a different RKHS (this is not statistical in nature in anymore).

Firstly, in my opinion, neither of them is inherently better than the other. The better model usually depends on the application at hand. Moreover, this modelling assumption is not a "remedy" for Gaussianity or iid-ness of noise. This seems to have been suggested implicitly in the text, which is something I don't think is correct. This is not a dealbreaker but an important point to keep in mind.

Secondly, modelling the noise as an RKHS function is unusual. Is there some inherent motivation as to why this is a good model to capture the underlying process? In particular, this is non-statistical model which is a very uncommon method to account for noise. Is there a particular reason for which the authors use this as a part of their model, apart from analytical tractability? Are there specific applications for which such a model would be a reasonably good approximation of the true underlying process?

Thirdly, the term "energy-boundedness" is thrown pretty loosely throughout the paper. The energy of a function or a signal is usually measured in its L2 norm, not the RKHS norm. While these ideas coincide when the noise kernel is taken to be the dirac delta, this is not the case in general.  Once again, this modelling assumption seems uncommon. Can the authors justify this particular choice beyond the fact that is makes the problem tractable. More importantly, a bounded L2 norm implies a bounded RKHS norm but the converse does not hold. Thus, this is a more stringent assumption than the standard interpretation of energy-boundedness.

Fourthly, the authors make the following claim:
> we derive an analytical solution that exactly characterizes the worst-case realization within the function hypothesis class

If I understand this correctly, the authors are referring to Eq.8, the statement of Lemma 1. If the authors are referring to $\overline{f}^{\sigma}(\cdot)$ as "the worst-case realization within the function hypothesis class", then the statement is incorrect because it does not belong to RKHS. In particular $\Sigma_{\sigma}^{f}$ is not an element of the RKHS and hence neither is $\overline{f}^{\sigma}(\cdot)$. I am not saying that Lemma 1 is incorrect. All I am saying is that you are not obtaining "an optimal f", but an optimal value of (an upper bound on) "$f(x_{N+1})$", a scalar. These two are different things.


I am also curious about the actual benefit of using this approach in downstream tasks. Let us compare the uncertainty estimates from the classical iid Gaussian and the proposed methodology in this work (the comparison is not really apple to apple, but let us assume for the moment). Say the idealized optimal bounds in this work are tighter than those produced by the Gaussian assumption. However, there is no quantification of the improvement over the classical conservative bounds. Moreover, in practice the optimal value of $\sigma$ will not be known and will be estimated using some sort of iterative procedure, which will worsen the estimates obtained from this method. Furthermore, optimizing over $\sigma$ is computationally expensive as calculating a single gradient requires $\mathcal{O}(N^3)$ operations. Thus, despite the increased computational requirements it is not clear how much benefit, if any, is obtained using such an approach. I understand that carrying out such experiments is beyond the scope of this work and I don't expect it to be a part of this paper either. However, without any quantification of improvement in uncertainty estimates, the benefit of such an approach remains unclear.

Lastly, how sensitive is the proposed approach to the choice of the RKHS, especially for the noise? If the estimates vary wildly depending on the choice of RKHS, then such a methodology is not practically robust.


My current score is more reflective of the questions above rather than the core technical content of the paper. I am willing to reconsider it if the authors can provide answers to my questions above.

**Ethical Concerns:**

["NO or VERY MINOR ethics concerns only"]

**Final Justification:**

Please refer to my comment to the author's response.

**Limitations:**

Yes

**Quality:**

2

**Strengths And Weaknesses:**

I think the paper is decent. It seems to be technically sound and has some new ideas in there. My main concern with the paper is regarding the modelling aspect, which I have explained in the section below.

---

> ### Author Rebuttal · Authors · 2025-07-30
>
> We thank the Reviewer for the careful reading of the paper.\
> **On modeling noise sequences as members of deterministic function
> spaces:** We agree that the choice of giving a stochastic or
> deterministic representation of the noise depends on the problem at
> hand. There are indeed situations in which stochastic modeling is
> preferable -- e.g., when samples are i.i.d. with known distribution.
> However, the choice of modeling noise sequences as energy-bounded
> members of a RKHS (which is also performed in \[Kanagawa+2018\]) allows
> us to overcome two main difficulties of stochastic representations,
> namely dealing with biases and correlation among samples. Additionally,
> such a set-up allows us to deal with adversarial noises (in fact, we are
> computing the bound accounting for the worst-case noise realization),
> and rich-enough hypotheses classes for the noise allow us to compensate
> for model misspecification of the latent function. Therefore, the
> presented modeling choice has practical advantages that go beyond
> tractability. Finally, we would like to point out a further connection
> between our set-up and the stochastic one (cf. footnote 2, p.8): if one
> considers sub-Gaussian and i.i.d. noise, it is possible to find a
> certain probability level for which the norm-bounds in Assumption 1 hold
> -- but the converse is not true.\
> **On energy-boundedness:** While the reviewer is correct that an
> RKHS-norm assumption is significantly more restrictive than a bound on
> the $\mathscr{L}^2$ norm, we would like to clarify that we are never
> invoking energy bounds or $\mathscr{L}^2$ norms on continuous functions.
> When introducing the RKHS condition in Assumption 1, we directly
> reference Section 4.1 regarding its interpretation as energy-bounded
> noise. In particular, for Dirac noise kernels, Equation (16) in
> Section 4.1 shows that the condition is equivalent to bounded energy of
> the finite noise sequence affecting the measurements, i.e.,
> $\sum_{i=1}^N w_i^2 \leq \Gamma_w^2$. Such energy-bounded noise
> assumptions are common in classical robust regression works such as
> \[Fogel 1979\], \[Bertsekas+2003\] and many others. We see that the word
> "energy\" might also be interpreted as a $\mathscr{L}^2$-energy bound
> and we plan to use a more precise wording to highlight that this is a
> bound on the finite and discrete noise sequence that affects our finite
> and discrete measured data $(x_i,y_i)$, $i=1,\dots N$. Our general
> theoretical results are valid for any positive definite and uniformly
> bounded kernel $k^w$ characterizing the noise. This general modeling
> framework ensures that we can simultaneously recover results from
> classical energy bounded regression \[Fogel 1979\], connect our results
> to to recent kernel error bounds \[Kanagawa+2018\], and provide an
> interpretation of the bound in terms of mean and covariance of a GP (cf.
> Section 4).\
> **On the claim in the contributions:** As the Reviewer correctly points
> out, our sentence "we derive an analytical solution that exactly
> characterizes the worst-case realization within the function hypothesis
> class\" is not precise. Indeed, given the non-parametric nature of the
> problem, the worst-case function $f^\star$ depends on the query point
> $x_{N+1}$ -- thus, the whole function $\bar{f}(x_{N+1})$, which takes
> the evaluation of different worst-case functions at different query
> points, is no longer in the hypothesis space. What we mean is that, for
> any given query point $x_{N+1}$, our procedure returns a worst-case
> latent function $f^\star$ in the RKHS space, cf. its parametrization in
> terms of the weight vector $\theta^\star$ in Eqs. (C.3), (C.16) for
> Cases 1 and 2. The evaluation of this function at the point $x_{N+1}$
> then returns the error bound $\bar{f}(x_{N+1})$. We plan to better
> convey this idea by moving some of the expressions of the worst-case
> function $f^\star$ from the Appendix to the main text.\
> **On the benefits of the proposed approach and application on a
> downstream task:** First, we would like to emphasize that if the
> distribution of the noise is known, probabilistic bounds are the most
> natural choice to quantify the uncertainty -- and this would be the case
> for Gaussian i.i.d. noise. However, as soon as the distribution is
> unknown (e.g., biased and correlated), applying GP bounds would return
> wrong results, possibly endangering safety for downstream tasks. This
> displays one of the strengths of the proposed approach: it is
> distribution-free, and is also capable of dealing with adversarial
> noises. We defer to footnote 2 for a discussion on the connection
> between the proposed bound and the probabilistic ones, but we plan on
> expanding on this point in the paper.\
> Regarding the optimization of $\sigma$, we highlight that this is only
> needed to obtain the least conservative error bound, while *any* value
> $\sigma$ provides a valid estimate that can be safely used for
> downstream tasks, as discussed at the end of Section 3.2. In other
> words, optimizing $\sigma$ leads to a refinement of the uncertainty
> envelope, and the termination of iterative optimization procedures can
> be dictated by the downstream task at hand.\
> To highlight these aspects and also display the performance of the
> proposed approach, we conducted an experiment where we apply the
> proposed bounds to a downstream task. Specifically, we consider
> safety-critical control of an uncertain dynamical systems, where we
> demonstrate significant benefits of the reduced conservatism and
> manageable computation times when jointly optimizing $\sigma$.
> Additionally, we highlight another key property of the proposed bound --
> they still hold when considering a subset of data -- which has a
> significantly bigger impact on the computational demand compare to the
> scalar $\sigma$.\
> *Experimental details:* We compare the proposed robust bound and a
> probabilistic bound akin to the setup in Section 4.3. The downstream
> application of the bound is thereby given by an optimization-based
> controller (specifically, a control barrier function) that ensures
> safety (constraint satisfaction) of an unknown dynamical system modeled
> using kernel regression with $10^2$ data points. The only difference in
> the application of the probabilistic bound, denoted by (a), compared to
> the proposed robust bound, (b), is the simultaneous optimization over
> the noise parameter $\sigma$ (one additional optimization variable), as
> well as the different scaling factor $\beta_\sigma$ of the covariance
> matrix, see also the discussion in Section 4.3. Additionally, we applied
> the proposed bound to a subset of data containing the 10 nearest
> training input locations and we denote this approach by (c). Notably,
> the derived robust bounds remain valid under arbitrary data selection
> rules, while the probabilistic approach relies on martingale bounds,
> which would fail under such data-selection rules. The following table
> compares the three approaches in terms of their computation time to
> solve the optimization problem to convergence, as well as the
> conservativeness of the bound, indicated by the share of the input
> domain for which a safety-preserving (feasible) input is computed:
>
>  |                             |     (a)    |    (b)  |     (c) |
>  | ----------------------- | ------- | ------- | -------  |
>  |    Time / iter. \[ms\]  |   5.7  |  17.9  |  0.4 |
>  |  Feasible domain \[%\]  |  63  |    81   |   62 |
>
> The larger feasible domain of the proposed bound (b) illustrates the
> reduced conservatism of the bound. While, for the same number of data
> points, the computation time for optimizing the robust bound (b) is
> visibly larger than for the probabilistic one (a), the subset selection
> mechanism (c) leads to a similarly large feasible domain as the
> probabilistic bound at significantly reduced computation times.\
> **On the sensitivity of the bounds to the choice of the RKHS:** Choosing
> the RKHS is a widely investigated topic in the statistical learning
> literature, and connects with results obtained in the realm of Gaussian
> process regression, cf. \[Rasmussen+2006, Sec. 5.4\]. Our set-up falls
> also into this framework: indeed, a key result of our work is that the
> worst-case error bound has the same formula as the error bound for
> standard Gaussian process regression for a suitably chosen noise
> $\sigma$ when choosing a Dirac kernel for the noise, see Equation (8).
> Although less prevalent, we note that also general (non-Dirac) noise
> kernels $k^w$ are leveraged for Gaussian process regression
> \[Rasmussen+2006, Sec. 5.4.3\] and the equivalence (Eq. (8)) equally
> applies in this case. Hence, the sensitivity of our bound to the choice
> of the RKHS is comparable to the sensitivity of general Gaussian process
> regression to the choice of the kernel. For this reason, we see no need
> to separately investigate the sensitivity to related hyperparameters.
>
> **References:**\
>  \[Kanagawa+2018\] M. Kanagawa, P. Hennig, D. Sejdinovic, and B. K.
> Sriperumbudur, "Gaussian Processes and Kernel Methods: A Review on
> Connections and Equivalences," arXiv:1807.02582, 2018;\
>  \[Fogel 1979\] E. Fogel, "System identification via membership set
> constraints with energy constrained noise," IEEE Transactions on
> Automatic Control, vol. 24, no. 5, pp. 752--758, 1979;\
>  \[Bertsekas+2003\]: D. Bertsekas and I. Rhodes, "Recursive state
> estimation for a set-membership description of uncertainty," IEEE
> Transactions on Automatic Control, vol. 16, no. 2, pp. 117--128, 1971;\
>  \[Rasmussen+2006\]: C. E. Rasmussen and C. K. I. Williams, Gaussian
> processes for machine learning. in Adaptive computation and machine
> learning. Cambridge, Massachusetts: MIT Press, 2006.

---

> > ### Comment · Reviewer_BU3a · 2025-08-06
> > **Response to Authors**
> >
> > Thanks for your detailed reply.
> >
> > On modelling assumptions: I agree that current analysis cannot handle biased and correlated noises. However, whenever you mention bias and correlation, it implicitly implies a statistical nature as these are statistical quantities. This is at odds with the deterministic setup in this work. Yet again, none of them is clearly better, but the comparison is still not apples to apples. With that said, I would want to add that bias and correlation can be taken of in a relative straightforward manner at least for cases when the query points (the $x$'s) are independent of the noise sequence. I agree we would need to some correlation structure but that somewhat comparable to knowing the RKHS here. Coming to the claim of adversarial noise, I understand what the authors are hinting at here. I agree with the general sentiment, but without a model of what "adversarial noise" refers to, particularly in context of the power the adversary has, the claim is more an appeal to our intuition rather than a concrete, provable statement. I think this is turning into a more of philosophical discussion at this point rather than a technical one. As a finishing comment, I would encourage the authors to include further motivation along the lines of our discussion here, outlining more about the adopted modelling choice and its pros and cons.
> >
> > On energy-boundedness: Once again, I understand the sentiment here but as mentioned earlier, energy of a function commonly refers to its L2 norm. I think a more precise characterization here would benefit both the paper and the reader.
> >
> > On downstream tasks: Thanks a lot for the added experiment. It definitely helps build the case for the paper.
> >
> > On sensitivity of bounds to choice of RKHS: I agree that choosing the RKHS is a widely investigated topic in the statistical learning literature. However, the key point to note here is that this is a choice for the underlying function, not the noise. When you also a noise component which belongs to RKHS, there are now two free choices of RKHS and choosing the correct one and their corresponding parameters is twice as hard. As a result, I don't think this is comparable to the problem in existing studies and demands a certain investigation or atleast a high-level sensitivity analysis.
> >
> >
> > Overall, I feel the author's response has helped improve the paper in my opinion and I have accordingly raised my score assuming the authors will update the paper as discussed.

---

> > > ### Author Response · Authors · 2025-08-07
> > >
> > > Thank you for your response and for the insightful discussion.
> > > We are happy to detail the discussion of the noise modeling assumption and choice of RKHS bounds, as well as improve clarity in terms of nomenclature (i.e. energy-boundedness of the noise sequence).

---

### Official Review · Reviewer_Ep7a · 2025-06-26

**Clarity:** 3
**Significance:** 3
**Originality:** 3
**Rating:** 5
**Confidence:** 3

**Summary:**

This paper presents tight non-asymptotic uncertainty bounds for kernel-based regression under the assumption of energy-bounded noise.   In this setting, no distribution is assumed on the noise, nor any independence assumptions.   Both the latent function being recovered and the noise are modeled as elements in reproducing kernel hilbert spaces (RKHS).

The authors formulate the problem as an infinite dimensional optimisation problem to characterise the worst case deviation of the function at a single arbitrary evaluation point.   The very neat idea, is that by relaxing / collapsing two inequality constraints, the problem can be simplified into one with a closed form solution, which is very nice.   The exact bound can then be recovered by optimising the solution over the scalar noise parameter.   The approach generalises and recovers classical results in kernel interpolation and linear regression, and is shown empirically to yield less conservative uncertainty regions than standard probabilistic bounds in low data-regimes.

**Questions:**

1. How sensitive are your results to mis-specification of the rkhs norm bounds?   Could you provide some examples of how they can be estimated from data, and how estimation error affects your bounds?

2. Can some details be provided about how the optimisation in (9) can be carried out?   Does this run into stability issues as $\sigma\rightarrow 0$?  If so, is some regularisation needed?

3. Is it possible to demonstrate how these bounds are used in a down-stream setting?  e.g. Bayesian optimisation or similar applications?

**Ethical Concerns:**

["NO or VERY MINOR ethics concerns only"]

**Final Justification:**

The authors have addressed all my questions, and I appreciate their efforts in adding a good downstream example.  I have updated my score accordingly.

**Limitations:**

yes

**Quality:**

3

**Strengths And Weaknesses:**

Strengths:
1. This approach provides a tight, non-asymptotic bounds for kernel regression that handles noise under no distributional assumptions.
2. The bounds derived are straightforward to recover through the relaxation approach.
3. Various classical results are obtained as a special case, and some novel bounds are obtained in the probabilistic case.


Challenges with this approach:
1. The main challenge is the assumption of a known RKHS bound of both the function and the noise.    In practice this would be approached empirically which would influence the derived bounds significantly - as mentioned in the paper as an acknowledged limitation.
2. The method does not appear to lend itself well to bounds over multiple prediction points.   It is not clear to me that it would be straightforward to extend the approachd directly to ths setting.
3. It would have been nice for the authors to add a demonstration of how these bounds can be effectively deployed in downstream tasks.

---

> ### Author Rebuttal · Authors · 2025-07-30
>
> We thank the Reviewer for the careful reading of the paper.\
> **On knowing the RKHS norm bound:** This assumption is common with other
> state-of-the-art bounds (e.g.
> \[Srinivas+2012,Chowdury+2017,Fiedler+2021\]) that have been successful
> in downstream tasks such as safe reinforcement learning and Bayesian
> optimization. Given the practical impact of such results, there exist
> works that estimate RKHS norms: see e.g. \[Tokmak+2025\] and references
> therein for extrapolation-based approaches, and \[Csaji+2025\] for the
> case of Paley-Wiener kernels of band-limited functions using the
> uniformly-randomized Hoeffding's inequality or the empirical Bernstein
> inequality. Another strategy for estimating the RKHS bounds can be
> developed by expanding on ideas from the robust regression literature,
> such as \[Calliess+2020\]: one starts from the smallest bound that is
> not falsified by the data (i.e., such that the problem remains feasible)
> and increases it with a small proportional buffer according to the
> so-called "lazy adaptation\" technique; then, such a bound can be
> combined with recent RKHS-norm extrapolation heuristics \[Tokmak+2025\].
> However, a formal analysis of this technique to the considered setting
> and intensive numerical validations would be required. We plan to expand
> on this point in the limitation section of the paper.\
> **Sensitivity of result to norm-bound misspecification:** A thorough
> quantitative analysis depends on the adopted estimation strategy and
> will be deferred to future works. Nevertheless, we can mention the
> following cases:
>
> -   if the norm bounds are too small, this could be detected by checking
>     feasibility of the constraints in the problem formulation (C.1);
>
> -   if the problem (2) remains feasible despite $\Gamma_f$ being too
>     small, then the noise function given by a positive definite noise
>     kernel $k^w$ can generally compensate for the inadequacy of the
>     constraint on the latent function norm.
>
> Finally, we would like to point out that taking conservative (too large)
> norm bounds for both the noise and the latent function does not have a
> dramatic impact on the resulting bound. In fact, as displayed in
> Equation (7), conservative values for $\Gamma_f^2$ and $\Gamma_w^2$ lead
> to a just sublinear inflation of the uncertainty envelope.\
> **Validity of proposed bound over multiple prediction points:** Our
> deterministic guarantees are derived for any query point $x_{N+1}$ given
> the training data -- this is also what is done in our experiments
> (cf. Figure 1 and 2), where the bound is computed for any input in the
> given domain.\
> **Deployment in downstream tasks:** We agree that quantifying the
> performance of the proposed approach in a downstream task would better
> highlight some of the benefits. Therefore, we conducted the following
> experiment. We are given a dynamical system with unknown (scalar)
> dynamics, which is estimated from $10^2$ data points using kernel-based
> estimation. The estimated dynamic model and its uncertainty bound is
> utilized in a control barrier function (CBF) to ensure that the
> optimized control input satisfies safety-critical constraints. Through
> the size of the uncertainty envelope, the conservatism of the employed
> bound determines the feasible domain for which safe inputs can be
> computed, with smaller confidence sets leading to a larger feasible
> region. Similar to the numerical comparison in Section 4.3, we compare
> three methods for computing the uncertainty bound: (a) the probabilistic
> bound by \[Abbasi-Yadkori 2013\]; (b) the derived robust regression; (c)
> the derived robust regression using only the closest $10^1$ data points
> for each test point. Apart from the number of data points used, the
> corresponding optimization problems only differ in the number of
> optimization variables (for the robust bounds, we additionally optimize
> over the free noise parameter $\sigma$), as well as the definition of
> the scaling factor $\beta$, cf. Eq. (7) for the robust case and
> \[Fiedler+2024, Eq. (7)\] for the probabilistic case. Crucially,
> benchmark (c) also highlights the fact that the derived bound remains
> valid for any subset of data, independent of how it is chosen (in
> contrast to stochastic error bounds). The following table compares the
> robust and probabilistic bounds in terms of the computational
> complexity, as well as their conservatism in terms of the portion of the
> domain for which the size of the bound retains feasibility of the
> optimization problem:
>
>  |                             |     (a)    |    (b)  |     (c) |
>  | ----------------------- | ------- | ------- | -------  |
>  |    Time / iter. \[ms\]  |   5.7  |  17.9  |  0.4 |
>  |  Feasible domain \[%\]  |  63  |    81   |   62 |
>
>
> While optimization over the noise parameter in (b) leads to larger
> computation times for the same number of data points, it leads to a
> larger feasible domain due to its reduced conservatism even in the
> zero-mean i.i.d. setting. The subset-of-data approach (c) highlights
> that the method can be computationally more efficient, while also
> providing rigorous guarantees for safety-critical applicants in
> downstream tasks.\
> **On solving (9):** The main difficulty when optimizing (9) lies in the
> fact that the optimizer $\sigma$ can tend towards the boundaries of
> $(0, \infty)$, which can be addressed numerically by using adequately
> permissive box constraints, for example by enforcing
> $\sigma^2 \in [10^{-6}, 10^6]$. To avoid scaling issues, in our
> experiments we have found that optimizing over $\log(\sigma^2)$ improves
> numerical stability; alternatively, squashing functions can be employed
> to optimize in the large parameter space. Additionally, we have added a
> small constant "jitter" $\sigma_{\mathrm{jitter}}^2 = 10^{-6}$ to the
> kernel matrix, i.e., used
> $K^f + (\sigma^2 + \sigma_{\mathrm{jitter}}^2) I$ in the implementation,
> as is commonly done to avoid numerically indefinite kernel matrices.
> Using these techniques, we have not experienced numerical issues in the
> limiting case $\sigma \rightarrow 0$, which we observed in our
> experiments for some query input locations $x_{N+1}$ that are close to a
> training input location. This is expected since, as shown in
> Proposition 2, the optimized bound attains a finite value for
> $\sigma \rightarrow 0$.
>
> **References:**\
>  \[Srinivas+2012\] N. Srinivas, A. Krause, S. M. Kakade, and M. W.
> Seeger, "Information-Theoretic Regret Bounds for Gaussian Process
> Optimization in the Bandit Setting," IEEE Transactions on Information
> Theory, vol. 58, no. 5, pp. 3250--3265, 2012;\
>  \[Chowdury+2017\] S. R. Chowdhury and A. Gopalan, "On Kernelized
> Multi-armed Bandits," in Proceedings of the 34th International
> Conference on Machine Learning, PMLR, 2017, pp. 844--853;\
>  \[Fiedler+2021\] C. Fiedler, C. W. Scherer, and S. Trimpe, "Practical
> and Rigorous Uncertainty Bounds for Gaussian Process Regression,"
> Proceedings of the AAAI Conference on Artificial Intelligence, vol. 35,
> no. 8, Art. no. 8, 2021;\
>  \[Tokmak+2025\] A. Tokmak, K. G. Krishnan, T. B. Schön, and D. Baumann,
> "Safe exploration in reproducing kernel Hilbert spaces", 2025, arXiv:
> arXiv:2503.10352;\
>  \[Csaji+2025\] B. C. Csáji and B. Horváth, "Derandomizing Simultaneous
> Confidence Regions for Band-Limited Functions by Improved Norm Bounds
> and Majority-Voting Schemes", 2025, arXiv: arXiv:2506.17764;\
>  \[Calliess+2020\]: J.-P. Calliess, S. J. Roberts, C. E. Rasmussen, and
> J. Maciejowski, "Lazily Adapted Constant Kinky Inference for
> nonparametric regression and model-reference adaptive control,"
> Automatica, vol. 122, p. 109216, 2020;\
>  \[Fiedler+2024\]: C. Fiedler, J. Menn, L. Kreisköther, and S. Trimpe,
> "On Safety in Safe Bayesian Optimization", 2024, arXiv:
> arXiv:2403.12948;\
>  \[Abbasi-Yadkori 2013\]: Y. Abbasi-Yadkori, "Online learning for
> linearly parametrized control problems," University of Alberta,
> Edmonton, Alberta, 2013.

---

> ### Comment · Reviewer_Ep7a · 2025-08-05
> **Thanks**
>
> I thank the authors for their answers.    The sublinear dependence on the norms in (7) is a key point which the authors point out, which does mitigate the issue of choosing these bounds, to some extent.
>
> Regarding the multiple prediction points - yes, of course, the paper is about recovery of the latent function through regression.  I was thinking more in the context of uncertainty (e.g. GP) where the predictive posterior across multiple points is correlated.    I see this is not relevant here.
>
> The dynamic system numerical experiment would be a very nice addition to the paper - thanks for adding this.
>
> Thanks for addressing my questions.

---

### Official Review · Reviewer_sCCF · 2025-06-28

**Clarity:** 3
**Significance:** 3
**Originality:** 3
**Rating:** 5
**Confidence:** 3

**Summary:**

This paper introduces a novel framework for deriving tight, non-asymptotic uncertainty bounds in kernel-based regression under energy-bounded noise, formulated as a deterministic optimization problem. The proposed method generalizes existing kernel interpolation and linear regression results, offering a robust alternative to probabilistic approaches. Theoretical derivations are rigorous, and the connection to Gaussian process (GP) regression and RKHS theory is well-structured. The bounds are computationally efficient, relying on scalar optimization, and demonstrate advantages in low-data regimes.

**Questions:**

1.	What practical strategies (e.g., cross-validation, online adaptation) could be employed to estimate the latent function complexity and noise energy from data?
2.	Under what conditions (kernel choices, noise structures) can we guarantee convexity in the σ optimization problem?
3.	How sensitive are the bounds to parameter misspecification, particularly when the true function lies outside the assumed RKHS?

**Ethical Concerns:**

["NO or VERY MINOR ethics concerns only"]

**Final Justification:**

As my rating is already positive, I would like to keep my current rating.

**Limitations:**

The authors do discuss the limitations including challenges of kernel mis-specification, estimating the RKHS-norm bound, and handling large datasets in their work.

**Paper Formatting Concerns:**

No.

**Quality:**

3

**Strengths And Weaknesses:**

Strengths
1.	The use of RKHS-norm-bounded noise provides a deterministic framework that relaxes traditional i.i.d. or Gaussian noise assumptions, addressing a gap in the literature. The derived bounds are tight and non-asymptotic, with elegant connections to GP posterior variance.
2.	The reliance on scalar optimization makes the approach computationally tractable and suitable for iterative refinement in practical applications like safe reinforcement learning and Bayesian optimization.
3.	As reported, empirical results demonstrate superior performance to probabilistic methods in low-data regimes.

Weaknesses
1.	The method's core premise is prior knowledge of latent function complexity and noise energy, which are challenging to estimate in practice. Sudden increases in noise energy (e.g., adversarial perturbations) may violate preset noise energy, compromising safety guarantees.
2.	Unlike probabilistic bounds, the proposed bounds do not improve with more data under energy-bounded noise. This trade-off between robustness and asymptotic efficiency requires deeper discussion.
3.	Comprehensive testing of parameter sensitivity and validation under noise correlation structures is lacking.

---

> ### Author Rebuttal · Authors · 2025-07-30
>
> We thank the Reviewer for the insightful comments.\
> **On the energy bounds and adversarial noises:** Assuming to have known
> RKHS-norm bound for the unknown latent function is quite standard in the
> literature \[Srinivas+2012,Chowdhury+2017,Fiedler+2021\] (see the
> paragraph "Practical strategies to estimate RKHS-norm bounds\" below for
> further discussion and existing techniques to estimate the RKHS norm).
> Therefore, given adequate RKHS-norm bounds, our approach is capable of
> addressing the case of adversarial noise, possibly with non-zero mean:
> by construction, the proposed bound accounts for the worst-case
> adversarial noise, regardless of its distribution. Note that this is
> typically not the case for standard regression results, which leverage
> the assumption that noises are zero-mean i.i.d. and cannot handle
> adversarial noise.\
> **Discussion on the performance as the data-set size increases:** We
> would like to clarify that the derived bound is tight, i.e., given the
> posed assumptions, it provides the exact description of uncertainty
> envelope for the latent function given the data. Under the assumption
> that the noise is zero-mean i.i.d., one can usually guarantee to learn
> the unknown function up to any tolerance. With the proposed method, we
> can also generate monotonically improving estimates of the unknown
> function; however, due to the adversarial noise assumption, there is no
> guarantee that the bounds will asymptotically shrink to zero (cf. Figure
> 2 in the paper). Note that standard stochastic regression bounds working
> with i.i.d. and zero-mean noises cannot handle adversarial (or just
> slightly biased) noises, and their well-known behavior of returning
> vanishingly small uncertainty bounds would lead to an incorrect bound in
> such a set-up.\
> **Testing of parameter sensitivity and validation under noise
> correlation:** Modeling noise sequences as energy-bounded members of
> some deterministic function space allows us *by construction* to handle
> biased and correlated noise. An informed choice of the noise kernel can
> thereby lead to improved estimates; in particular, by using the Dirac
> kernel $k^w$, the derived bounds are valid for any noise sequence
> satisfying $\sum_{i=1}^N w_i^2< \Gamma_w^2$ (see Equation (16)), even if
> the noise is correlated.\
> **Practical strategies to estimate RKHS-norm bounds:** Many
> state-of-the-art uncertainty bounds for kernel-based methods rely on
> RKHS-norm bounds \[Srinivas+2012,Chowdury+2017,Fiedler2021\], which are
> of high relevance in downstream tasks, such as safe reinforcement
> learning \[Berkenkamp+2023\] and Bayesian optimization \[Sui+2018\].
> This has motivated works on estimating the RKHS-norm bound. For
> instance, \[Tokmak+2025\] and its preceding works elaborate an
> extrapolation-based technique to estimate such a quantity, and
> \[Csaji+2025\] derives bounds for Paley-Wiener kernels of band-limited
> functions using a uniformly-randomized Hoeffding's inequality or an
> empirical version of Bernstein's inequality. Additionally, one would
> also be capable of detecting the inadequacy of the norm bounds (i.e.,
> them being too small) by checking feasibility of the constraints in the
> formulation given in Equation (C.1): this would allow to rule out
> certain configurations, as they cannot explain the data.\
> **What conditions would yield convexity in the $\sigma$-optimization
> problem?** This is a very interesting point. We would like to highlight
> that we never claim that the problem is convex and application of the
> derived bound to downstream tasks also does not require this. Instead,
> the key point regarding Problem (9) is that any variable
> $\sigma\in(0,\infty)$ provides a correct error bound. Hence, for
> decision making, we can simply treat $(x_{N+1},\sigma)$ as a decision
> variable, and Theorem 1 ensures that we always obtain correct error
> estimates and that the optimal choice of $\sigma$ also ensures that this
> bound is sharp. As both the ground truth function $f^{\mathrm{tr}}$ and
> the upper bound $\bar{f}^\sigma(x)$ are typically non-convex, convexity
> in $\sigma$ plays no crucial role. Moreover, convexity is not required
> to achieve global optimality using simple gradient descent algorithms.
> In this regard, we expect that Problem (9) satisfies the
> Polyak-Łojasiewicz (PL)-conditions, although we currently have no formal
> proof.\
> **Sensitivity of the bounds to parameter misspecification, especially
> when the RKHS does not contain the true function:** The proposed method
> can deal with the presented misspecification, because we do not treat
> the noise sequence as a stochastic object, but as a member of a
> deterministic function space. As long as the latter is rich enough, the
> noise is therefore capable of compensating for the misspecification of
> the latent-function RKHS. In particular, if data are generated by
> $y_i=h(x_i)$ with some function $h$ that is not in the RKHS, then we can
> decompose $h(x)=f(x)+w(x)$, where $f$ lies in the RKHS and $\Gamma_f$
> bounds its RKHS norm, while $w$ is the residual. By picking $k^w$ as the
> Dirac kernel, Assumption 1 holds with
> $\sum_{i=1}^N |h(x_i)-f(x_i)|^2<\Gamma_w^2$ (see Equation (16)): for the
> noise to "compensate" the RKHS misspecification, its bound on the RKHS
> norm of the noise has to be larger than the point-wise squared distance
> between the true function and the function that lies in the RKHS.
>
> **References:**\
>  \[Srinivas+2012\] N. Srinivas, A. Krause, S. M. Kakade, and M. W.
> Seeger, "Information-Theoretic Regret Bounds for Gaussian Process
> Optimization in the Bandit Setting," IEEE Transactions on Information
> Theory, vol. 58, no. 5, pp. 3250--3265, 2012;\
>  \[Chowdury+2017\] S. R. Chowdhury and A. Gopalan, "On Kernelized
> Multi-armed Bandits," in Proceedings of the 34th International
> Conference on Machine Learning, PMLR, 2017, pp. 844--853;\
>  \[Fiedler+2021\] C. Fiedler, C. W. Scherer, and S. Trimpe, "Practical
> and Rigorous Uncertainty Bounds for Gaussian Process Regression,"
> Proceedings of the AAAI Conference on Artificial Intelligence, vol. 35,
> no. 8, Art. no. 8, 2021;\
>  \[Tokmak+2025\] A. Tokmak, K. G. Krishnan, T. B. Schön, and D. Baumann,
> "Safe exploration in reproducing kernel Hilbert spaces", 2025, arXiv:
> arXiv:2503.10352;\
>  \[Csaji+2025\] B. C. Csáji and B. Horváth, "Derandomizing Simultaneous
> Confidence Regions for Band-Limited Functions by Improved Norm Bounds
> and Majority-Voting Schemes", 2025, arXiv: arXiv:2506.17764;
>  \[Berkenkamp+2023\]: F. Berkenkamp, A. Krause, and A. P. Schoellig,
> "Bayesian optimization with safety constraints: safe and automatic
> parameter tuning in robotics," Mach Learn, vol. 112, no. 10, pp.
> 3713--3747,2023;\
>  \[Sui+2018\]: Y. Sui, V. Zhuang, J. Burdick, and Y. Yue, "Stagewise
> Safe Bayesian Optimization with Gaussian Processes," in Proceedings of
> the 35th International Conference on Machine Learning, PMLR, 2018, pp.
> 4781--4789.

---

> > ### Comment · Reviewer_sCCF · 2025-08-04
> > **Thanks for the responses**
> >
> > While the authors addressed several points, key theoretical limitations persist:
> > (1) Feasibility checks cannot detect spurious decompositions (mathematically valid but physically implausible function-noise pairs satisfying constraints).
> > (2) Theoretical guarantees for σ-optimization are absent, with PL-conditions presented as an expectation rather than a guarantee.
> > (3) Kernel misspecification discussion is limited to Dirac kernels, lacking quantification of error propagation for practical kernels (e.g., Matérn, RBF).
> > Since my rating is already positive, I would like to keep my current rating.

---

> > > ### Author Response · Authors · 2025-08-04
> > >
> > > We thank the reviewer for the response.
> > >
> > > 1.  The reviewer is correct that the theoretical validity of proposed
> > >     method still relies on the RKHS assumption of the underlying
> > >     functions (Assumption 1), and validity of this Assumption 1 may be
> > >     hard to verify in experiments.
> > >
> > > 2.  We confirm that no theoretical guarantees are provided regarding the
> > >     convexity of optimization problem (9). In this regard, we would like
> > >     to highlight that the theoretical bound is valid for any $\sigma$,
> > >     ensuring safety in downstream tasks independent of convexity or PL
> > >     conditions.
> > >
> > > 3.  We would like to clarify that the above discussion for handling
> > >     kernel misspecification holds for any positive semi-definite
> > >     kernel $k^f$ of the latent function (including Matèrn and RBF
> > >     kernels). For the kernel of the noise $k^w$, compensation of the
> > >     misspecification in $k^f$ through the noise kernel requires that the
> > >     corresponding RKHS $\mathcal{H}_{k^w}$ allows for the noise to
> > >     interpolate the residual $h(x_i) - f(x_i)$ at the training input
> > >     locations -- an assumption that holds for any positive-definite
> > >     noise kernel (given a sufficiently large RKHS-norm bound
> > >     $\Gamma_w$), but provides a particularly easy-to-interpret condition
> > >     for the Dirac noise kernel.

---

### Official Review · Reviewer_fxeu · 2025-07-03

**Clarity:** 2
**Significance:** 2
**Originality:** 3
**Rating:** 4
**Confidence:** 3

**Summary:**

The paper studies kernel regression with both target and noise assumed to belong to certain RKHS's. The specific problem addressed is the maximum and minimum of the target at a new point given noisy observations on a training set and RKHS norm constraints for the target and noise. The paper shows that the solution can be obtained by solving a respective problem with a single linearly aggregated constraint. The paper particularly discusses two extreme special cases corresponding to permissive noise constraints and degenerate scenarios. Additionally, the paper shows that its results generalize earlier results on energy-bounded noise and noise-free interpolation, and compares its results to other deterministic and probabilistic bounds.

**Questions:**

I suggest to clarify for the reader the points I indicated above.

I suggest to provide the appendices in the same file after the bibliography; they are easier to access this way.

**Ethical Concerns:**

["NO or VERY MINOR ethics concerns only"]

**Final Justification:**

I don't have any significant concerns regarding this paper, and the authors have provided constructive responses to me and the other reviewers, so I have increased my score.

**Quality:**

3

**Strengths And Weaknesses:**

The paper is written at a high mathematical level, is generally quite readable, the authors are very familiar with the relevant literature and point out various connections of their results to earlier ones. The addressed problem is reasonable.

On the other hand, I'm not quite convinced in sufficient importance of the contribution, and there are some issues with the exposition.

line 91: *the optimization problem is infinite-dimensional and is not directly tractable.* - It is an exaggeration to call optimization problem (2) infinite-dimensional. It is obviously $(N+1)$-dimensional (or $2(N+1)$-dimensional): everything in the problem can be expressed in terms of suitable scalar products on the linear span of $N+1$ vectors corresponding to the measurement points $x_i$. The problem can be analyzed using an explicit finite-dimensional Lagrange function.

line 91: *Our key result, presented in the remainder of the section, consists in deriving an analytical solution for the bound evaluated at an arbitrary input location* - I only see analytic solutions for the two extreme cases $\\sigma=0$ and $\\sigma=\\infty$ in the main text. In Lemma 1 (and in the appendix) there is also an explicit formula for the general case $0<\\sigma<\\infty,$ but it depends on the unknown $\\sigma$, for which no explicit formula seems to be provided.

The main result (Theorem 1) asserts that the solution of the problem (2) can be obtained by minimizing the solutions of the relaxed problems with linearly aggregated constraints. This looks like a special case of a very general and well-known property of problems with linear objectives and convex constraints. Problem (2) has exactly this structure, as it can be written with a linear objective of the form $f\\mapsto\\langle v, f\\rangle$ and constraints $h_1(f)\\le 0, h_2(f)\\le 0$ with some convex quadratic polynomials $h_1,h_2$. Writing the Lagrange function and taking the gradient $\\nabla_f,$ we get $v-\\lambda_1 \\nabla_f h_1(f)-\\lambda_2 \\nabla_f h_2(f)=0.$  Assuming now, for example, that both constraints are active at the solution and $\\lambda_1,\\lambda_2>0,$ we get that the problem with the aggregated constraint $\\lambda_1 h_1(f)+\\lambda_2 h_2(f)\le 0$ has the same solution. There might be some subtleties related to degenerate cases, etc., but in any case Theorem 1 looks like a standard and expected result in constrained convex optimization, rather than a special feature of the RKHS optimization problem in question. This point is not properly clarified in the paper.

Since, as admitted by the authors themselves, the result in the special case $\\sigma=\\infty$ (effectively no noise constraint) is well-known, probably the most significant specific new result of the paper is the analysis of the opposite case $\\sigma=0$ associated with degenerate kernel matrices. Proposition 2 provides here an explicit answer. However, it's not very clear to me why this degenerate scenario is important or interesting.

It seems that Corollary 1 can alternatively be established without Proposition 2, just by replacing the objective $f(x_k)$ by $y_k-w(x_k)$ and using the $f$ - $w$ duality and Proposition 1.

---

> ### Author Rebuttal · Authors · 2025-07-30
>
> We thank the Reviewer for the insightful comments.\
> **On the claims in line 91:** We definitely agree that the problem
> stated in (2) becomes finite-dimensional after applying the Representer
> Theorem presented in Lemma A.1: nevertheless, the original problem is
> stated as searching for functions in infinite-dimensional spaces, and
> without exploiting the RKHS structure it might appear to be intractable.
> We plan to use a more precise wording to clarify this point. Also, we
> will make sure not to use \"analytical solution\" for the general
> outcome of Theorem 1: indeed, analytical (closed-form) solutions are
> available only for the two limit-cases $\sigma \to 0$ and
> $\sigma \to \infty$. The general result of Theorem 1, encompassing both
> limit cases $\sigma \to 0$ and $\sigma \to \infty$ as well as the
> scenario $0 < \sigma < \infty$, reduces (2) to a scalar, unconstrained
> optimization problem. This formulation is amenable for efficient
> iterative optimization and each iterate returns a valid (iteratively
> improving) bound for the unknown latent function.\
> **On Theorem 1 being standard in constrained convex optimization:** One
> of the core contributions of our work is to reduce the challenging
> problem of non-parametric function regression under non-i.i.d. noise to
> the analysis of a convex optimization problem. This solution is only
> obvious in hindsight, as we directly posed the problem in terms of an
> optimization problem in Equation (2). Existing results, especially
> \[Kanagawa+2018, Prop. 3.8\], also tried to address the same problem;
> however, their approach does not utilize a connection with convex
> optimization and, as we discuss in Section 4.2, fails to determine tight
> bounds for the robust regression problem. We note that the exact
> characterization of the optimal solution to this convex optimization
> problem is more involved than it may seem, as it cannot be simply
> assumed that both constraints are active -- instead, a careful handling
> of four different cases, corresponding to the possible sets of active
> constraints, is required. These are detailed in the supplementary
> material (Appendix C, pages 17-26).\
> **On the relevance of the case $\sigma \to 0$:** As we discuss in
> Section 3.3, degenerate kernel matrices leading to the case
> $\sigma\to 0$ occur in two practical situations:
>
> 1.  when the hypothesis space is determined by a finite set of $r<N$
>     basis functions, making it finite-dimensional and thus returning a
>     rank-deficient kernel matrix. This scenario occurs in many
>     approximations of kernel-based regression that aim at reducing its
>     $\mathcal{O}(N^3)$-computational complexity: these methods rely on
>     providing the eigen-decomposition of the kernel operator thanks to
>     Mercer's Theorem, and on considering a finite number of
>     eigenfunction as the basis (approximately) representing the
>     hypothesis space \[Zhu+1997\]; in the stochastic setting, a
>     paradigmatic approach leveraging this rationale in frequency domain
>     is that of \[Lázaro-Gredilla+2010\].
>
> 2.  when the test input $x_{N+1}$ belongs to the training data-set,
>     which can happen then evaluating the bound over the whole domain of
>     interest (cf. Figure 1 in the paper).
>
> Thus, the case $\sigma \to 0$ is also of practical relevance, and our
> analysis sheds some new light on regression with degenerate kernels.
> However, it is a byproduct of the general analysis in Theorem 1, which
> is capable of capturing and generalizing known results; see also the
> case $\sigma \to \infty$ and the discussion in Section 4.1 of the
> paper.\
> **On Corollary 1:** Thank you for this suggestion. In the current
> exposition, Corollary 1 follows directly from the more general claim of
> Proposition 2, which provides the analytic solution for the case
> $\sigma\to 0$ occurring when the kernel matrix is rank-deficient. From
> this perspective, the scenario in which the test-input belongs to the
> training data-set is one of the particular situations leading to a drop
> in the rank of the kernel matrix. However, we agree that Corollary 1
> could potentially be established analogously to Proposition 1 -- an
> insight that we are happy to convey in the paper.
>
> **References:**\
>  \[Zhu+1997\] H. Zhu, C. K. Williams, R. Rohwer, and M. Morciniec,
> "Gaussian regression and optimal finite dimensional linear models,"
> Aston University, NCRG/97/011, 1997;\
>  \[Lázaro-Gredilla+2010\] M. Lázaro-Gredilla, J. Quinonero-Candela, C.
> E. Rasmussen, and A. R. Figueiras-Vidal, "Sparse spectrum Gaussian
> process regression," The Journal of Machine Learning Research, vol. 11,
> pp. 1865--1881, 2010;\
>  \[Kanagawa+2018\] M. Kanagawa, P. Hennig, D. Sejdinovic, and B. K.
> Sriperumbudur, "Gaussian Processes and Kernel Methods: A Review on
> Connections and Equivalences," arXiv:1807.02582, 2018;\
>  \[Fogel 1979\] E. Fogel, "System identification via membership set
> constraints with energy constrained noise," IEEE Transactions on
> Automatic Control, vol. 24, no. 5, pp. 752--758, 1979.

---

> > ### Comment · Reviewer_fxeu · 2025-08-07
> >
> > I thank the authors for their response. I will increase my score.
> >
> > > We note that the exact characterization of the optimal solution to this convex optimization problem is more involved than it may seem, as it cannot be simply assumed that both constraints are active -- instead, a careful handling of four different cases, corresponding to the possible sets of active constraints, is required.
> >
> > Nevertheless, I suggest to more explicitly explain the connection to these general convex optimization ideas (while also commenting on the subtleties of their precise implementation). This would make the paper easier to understand.

---

> > > ### Author Response · Authors · 2025-08-07
> > >
> > > Thank you for your response and for the insightful discussion.
> > > We are happy to elaborate in the paper on connections to general concepts in convex optimization.

---

### Decision · Program_Chairs · 2025-09-17

**Decision:**

Accept (poster)

**Comment:**

The paper considers the worst-case bounds for kernel regression. It assumes the data are generated by the sum of two functions (a target function and a noise function) from a RKHS, and shows how to explicitly calculate bounds on the target function at a given out-of-sample point in terms of the RKHS norm bounds on the functions (assuming their respective kernels are bounded).

All reviewers liked it. They had some questions, all of which were suitable addressed by the rebuttals.

This is a theory paper, and NeurIPS has a lot of very strong theory papers and it cannot accept all of them. So I took a look, and I asked myself if the **setting** is interesting enough for a wide-range of NeurIPS readers. And I think it is: the setup seems very fundamental, and it also is an active area of research. I think kernel methods are one of the few areas of ML that have near state-of-the-art performance while still having guarantees; and similarly, GP regression and Bayesian Optimization are of interest to the typical NeurIPS reader. I don't know if this paper will have direct practical impacts, but it is making a theoretical contribution in an interesting area of ML.

Thus I'm pleased to recommend acceptance.